# TorchSpatial: A Location Encoding Framework and Benchmark for Spatial Representation Learning

**Nemin Wu**[1,3]\*,   **Qian Cao**[1,3]\*,   **Zhangyu Wang**[2,3],   **Zeping Liu**[3],   **Yanlin Qi**[3,4],
**Jielu Zhang**[1,3],   **Joshua Ni**[3,5],   **Xiaobai Yao**[1],   **Hongxu Ma**[6],   **Lan Mu**[1],
**Stefano Ermon**[7],   **Tanuja Ganu**[8],   **Akshay Nambi**[8],   **Ni Lao**[6,3†],   **Gengchen Mai**[3,1†]
[1]University of Georgia,   [2]UC Santa Barbara,   [3]SEAI Lab, University of Texas at Austin,
[4]UC Davis,   [5]Basis Independent Fremont,   [6]Google LLC,
[7]Stanford University,   [8]Microsoft Research,
{nemin.wu, qian.cao1, jielu.zhang, xyao, mulan}@uga.edu,
zhangyuwang@ucsb.edu,   zeping.liu@utexas.edu,   ylqi@ucdavis.edu,
nijoshua2025@gmail.com,   {hxma, nlao}@google.com, ermon@cs.stanford.edu,
{tanuja.ganu, akshay.nambi}@microsoft.com,   gengchen.mai@austin.utexas.edu
\*Equal contribution. Author ordering is determined by coin flip.   †Corresponding author.

## Abstract

Spatial representation learning (SRL) aims at learning general-purpose neural network representations from various types of spatial data (e.g., points, polylines, polygons, networks, images, etc.) in their native formats. Learning good spatial representations is a fundamental problem for various downstream applications such as species distribution modeling, weather forecasting, trajectory generation, geographic question answering, etc. Even though SRL has become the foundation of almost all geospatial artificial intelligence (GeoAI) research, we have not yet seen significant efforts to develop an extensive deep learning framework and benchmark to support SRL model development and evaluation. To fill this gap, we propose **TorchSpatial**, a learning framework and benchmark for location (point) encoding, which is one of the most fundamental data types of spatial representation learning. TorchSpatial contains three key components: 1) a unified **location encoding framework** that consolidates 15 commonly recognized location encoders, ensuring scalability and reproducibility of the implementations; 2) the **LocBench** benchmark tasks encompassing 7 geo-aware image classification and 10 geo-aware image regression datasets; 3) a comprehensive suite of **evaluation metrics** to quantify geo-aware models' overall performance as well as their geographic bias, with a novel **Geo-Bias Score** metric. Finally, we provide a detailed analysis and insights into the model performance and geographic bias of different location encoders. We believe TorchSpatial will foster future advancement of spatial representation learning and spatial fairness in GeoAI research. The TorchSpatial model framework and LocBench benchmark are available at `https://github.com/seai-lab/TorchSpatial`, and the Geo-Bias Score evaluation framework is available at `https://github.com/seai-lab/PyGBS`.

## 1   Introduction

Spatial representation learning (SRL) aims at learning neural spatial representations from spatial data (e.g., points, polylines, polygons, spatial networks, images, etc.) in their native formats while avoiding manual feature engineering [82, 81, 85] or data conversion (e.g., point clouds to voxels [58, 89, 14], polygons to raster images [6], or map vector files into raster image tiles [34, 24]). Learning good spatial representations is a fundamental problem and the key to achieving end-to-end

38th Conference on Neural Information Processing Systems (NeurIPS 2024) Track on Datasets and Benchmarks.

training for various GeoAI applications. However, several barriers are hindering the advancement of SRL research.

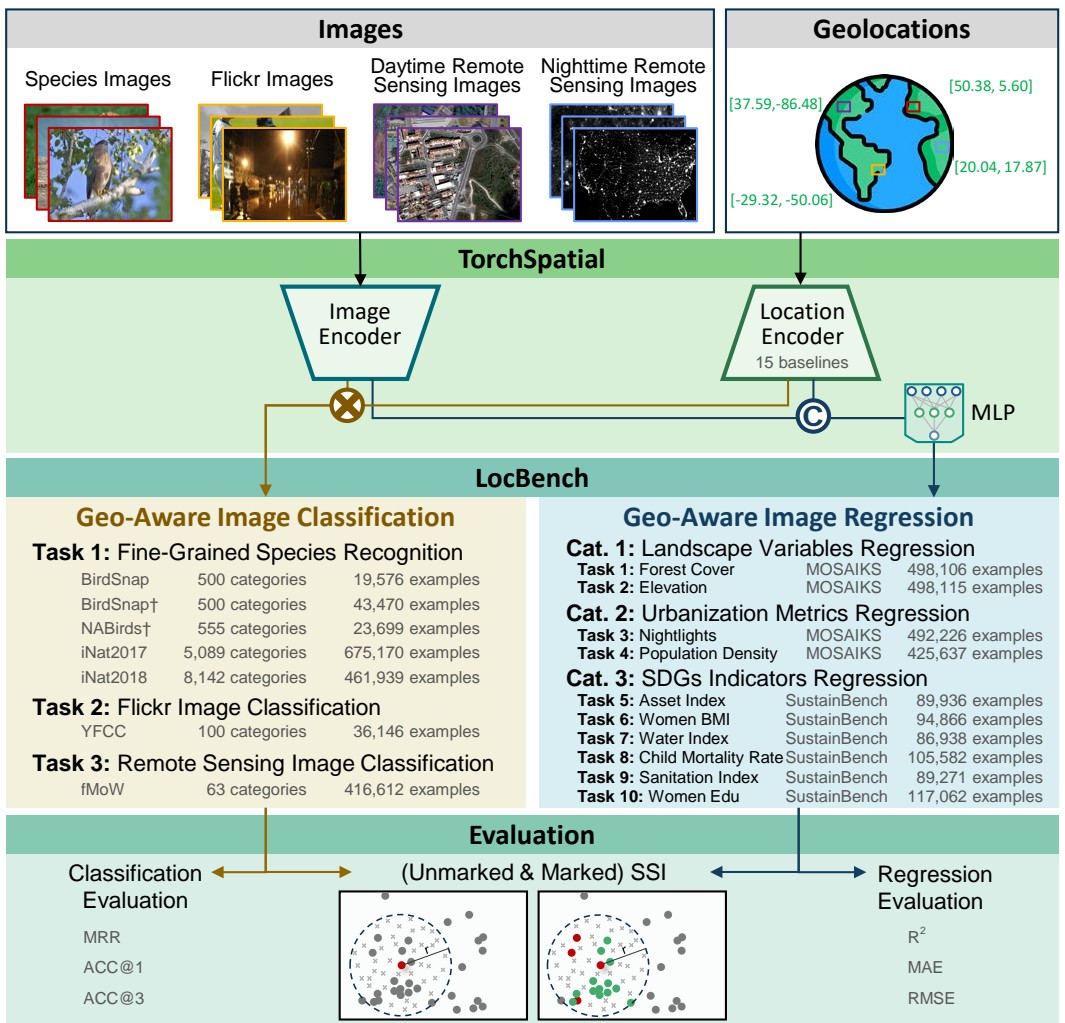

Figure 1: The overall framework of **TorchSpatial**. TorchSpatial provides a unified location encoding framework that consolidates 15 widely used location encoders and LocBench benchmark which contains 7 geo-aware image classification and 10 geo-aware image regression datasets. In addition, we provide a universally applicable geographic bias evaluation framework called Geo-Bias Score.

Firstly, there is no community-shared framework and benchmarks for SRL model development. A community-shared framework on a specific area can significantly accelerate research in that area. Examples are Torchvision[57] for computer vision tasks, TorchAudio[83] for audio and signal processing tasks, PyTorch Geometric [25] for graph neural network research, etc. While TorchGeo [69] has been developed for geospatial data processing and model development, it mainly focuses on processing geospatial image/raster data while much fewer efforts have been devoted to other geospatial data modalities (e.g., points, polylines, polygons, etc.) which are critical for GeoAI research. This significantly hinders GeoAI model development, as each research project must begin anew for much of the development (e.g., spatial data acquisition, processing, baseline reproduction, model development, and evaluation) without access to a standardized framework.

Secondly, location encoding of geolocation data [45, 18], one of the key components of SRL, has been proved useful for various geospatial tasks such as fine-grained species recognition [42, 51], satellite image classification [5, 53, 19, 35], weather forecasting [8, 37], and so on. However, no benchmark has been developed to systematically evaluate the location encoders' impact on model performance in tasks with diverse task setups, dataset sizes, and geographic coverage.

Last but not least, although many pioneering works demonstrated the effectiveness of geo-aware models [42, 46, 5, 47, 54, 51] in downstream tasks, there has not been work that systematically defines and evaluates the geographic bias of these geo-aware AI approaches (e.g., tile embeddings, location encoders, etc). The question of whether the additional geo-aware module mitigates or aggravates the geographic bias [39] has not been investigated. Generally speaking, spatial fairness and geographic bias research investigate whether learned AI models can perform equally well across geographic space. While these concepts have been proposed for a while, most efforts focus on the qualitative analysis of these biases of AI models such as large language models [23, 56], and almost no effort has been made to develop a universally applicable measure for such bias.

In this work, to fill these gaps, we present TorchSpatial, a deep learning framework and benchmark for spatial representation learning. Figure 1 illustrates the major components of TorchSpatial. The key contributions of TorchSpatial are threefold:

1. We provide a **TorchSpatial** model framework that supports location encoder development. Currently, TorchSpatial consolidates 15 widely used location encoders and necessary model building blocks for future location encoder development while ensuring scalability and reproducibility of the implementations.

2. We provide a **LocBench** benchmark which contains 7 geo-aware image classification and 10 image regression datasets. They are used to systematically evaluate the performance of any location encoder in datasets across varied task setups, geographic distributions, dataset sizes, etc.

3. We provide a comprehensive set of evaluation metrics to quantify location encoders' overall model performance as well as their geographic bias, with a novel **Geo-Bias Score**. To the best of our knowledge, this is the first universally applicable geographic bias evaluation framework designed to assess any AI models such as large language models [56, 55].

## 2 Related Work

**Spatial representation learning (SRL).** Spatial representation learning [52] aims at learning neural spatial representation of spatial data in their native format. According to the targeted spatial data types, SRL can be classified into location encoders [42, 46, 53, 51, 35, 18, 67, 13], polyline encoders [4, 29, 64, 86, 68, 65], polygon encoders [75, 33, 50, 80], polygon decoders [12, 1, 38, 87], etc. By automatically extracting a learning-friendly representation from different types of spatial data, SRL enables end-to-end training on top of spatial data. As one of the key components of SRL, location encoders aim at encoding a location into a learning-friendly representation that can be used in many downstream tasks such as fine-grained species recognition [42, 46, 53] and distribution modeling [18], population mapping [67], satellite image classification [51, 35], geographic question answering [44], etc. In this work, our TorchSpatial focuses on location encoder development and evaluation.

**Geo-Aware machine learning benchmarks.** There are many emerging benchmarks in machine learning that incorporate geographical information, particularly geographic coordinates. In earth observation, EarthNets[77] and GEO-Bench[36] integrate abundant Remote Sensing (RS) datasets for multiple domain-specific tasks, such as land cover classification, cloud segmentation, cattle counting, and RS change detection. In the ecology domain, the iNaturalist 2021 competition [32] provides a fine-grained species recognition dataset that includes images, their location metadata, and location uncertainty. Meanwhile, they also add location annotations to the previously released iNaturalist 2017[73] and iNaturalist 2018[31] datasets to motivate researchers leverage the spatial information effectively. Similarly, GeoLifeCLEF competition series [22, 11, 17, 40, 41, 10] provide a list of benchmark datasets for location-based species classification. In terms of image regression tasks, SustainBench[84] consists of 15 tasks across 7 Sustainable Development Goals. It provides datasets covering most countries in the Global South and certain Global North countries. MOSAIKS[66] is the largest benchmark dataset for RS image regression tasks to our knowledge, containing around 500,000 observations uniformly distributed around the globe, and it also proposes a CNN-based model for benchmarking. Despite the availability of many geo-aware machine learning benchmarks, many benchmarks such as EarthNets[77], GEO-Bench[36], SustainBench [84], and MOSAIKS[66] do not report performances of geo-aware models but only focusing on purely computer vision models even the location metadata is provided. Moreover, for benchmarks that emphasize the role of location information such as iNaturalist and GeoLifeCLEF, they follow a rather similar task setup. Our LocBench aims to systematically evaluate the impact of location encoders on model's overall performance and geographic bias across tasks with very different geographic coverage, dataset sizes, and task setups.

# 3 TorchSpatial Framework and Benchmark

## 3.1 TorchSpatial Model Framework

TorchSpatial is designed following the following framework proposed by [45]:

$$Enc(\mathbf{x}) = \mathbf{NN}(PE(\mathbf{x})), \tag{1}$$

Here $PE(\cdot)$ is a position encoder transforming location $\mathbf{x}$ into a $W$-dimension vector, namely position embedding. $\mathbf{NN}(\cdot) : \mathbb{R}^W \to \mathbb{R}^d$ is a learnable neural network module that maps the input position embedding $PE(\mathbf{x}) \in \mathbb{R}^W$ into the location embedding $Enc(\mathbf{x}) \in \mathbb{R}^d$. By following the common practice [71, 26, 78, 16, 42, 88, 46, 64], TorchSpatial framework is flexible enough to support the development of any location encoders. In the future, we plan also to support other spatial data types such as polylines, polygons, spatial networks, etc.

Within this framework, we implement 15 commonly recognized location encoders, and classify them into two groups: 1) 2D location encoders which work on a projected 2D space [7, 2, 46, 63, 42, 46] and 2) 3D location encoders which interpret geolocation as 3D coordinates [47, 60, 67]. Please refer to Appendix A.1 for a detailed description of these models.

For model inference, as depicted in Figure 1, there are two model inference structures tailored to the classification and regression tasks. Each structure is designed to leverage image and location data while aligning with the unique objectives of classification and regression.

For classification tasks, the objective is to predict discrete categories based on each input image and location, and the logits of the model are the possibility for each class. Inspired by [42], where location information is treated as a Bayesian spatio-temporal prior. In this setup, TorchSpatial separately processes the image and location data using two distinct classifiers: an image classifier and a location classifier. The final prediction is derived by performing an element-wise multiplication of the outputs from these two classifiers, as indicated by the brown circle with a cross in Figure 1. The influence of various location encoders can be assessed on classification accuracy while maintaining consistency in image representations. For regression tasks, the goal is to predict continuous numerical values. A more straightforward structure is adopted. Feature embeddings are extracted from both the image and location data using separate encoders. These embeddings are then concatenated to create a comprehensive representation, which is subsequently input into an MLP to predict continuous values. This process is illustrated by the dark blue circle labeled "C" for concatenation in Figure 1. The algorithm 1 presents the pseudocode for the model inference architecture described above.

---

**Algorithm 1** Pseudocode for Model Inference Architecture in TorchSpatial

---

1: **procedure** MODELINFERENCE($D_{type}$: dataset type; $Enc^{(I)}$: image encoder; $Enc^{(x)}$: location encoder; $(\mathbf{I}, \mathbf{x})$: an image and location tuple; $\mathbf{NN}$: MLP; $\odot$: element-wise multiplication; $[;]$: concatenation; softmax(): softmax function; $\hat{y}$: predicted variable)
2:     **if** $D_{type} == $ "classification" **then**
3:         $\mathbf{e}_I \leftarrow Enc^{(I)}(\mathbf{I})$
4:         $\mathbf{e}_x \leftarrow Enc^{(x)}(\mathbf{x})$
5:         $\hat{y} \leftarrow \text{softmax}(\mathbf{e}_I \odot \mathbf{e}_x)$
6:     **else if** $D_{type} == $ "regression" **then**
7:         $\mathbf{e}_I \leftarrow Enc^{(I)}(\mathbf{I})$
8:         $\mathbf{e}_x \leftarrow Enc^{(x)}(\mathbf{x})$
9:         $\mathbf{e}_{I,x} \leftarrow [\mathbf{e}_I; \mathbf{e}_x]$
10:        $\hat{y} \leftarrow \mathbf{NN}(\mathbf{e}_{I,x})$
11:     **end if**
12:     **return** $\hat{y}$
13: **end procedure**

---

## 3.2 LocBench

In order to systematically compare the location encoders' performance and their impact on the model's overall geographic bias, we clean and preprocess 7 geo-aware image classification datasets and 10 geo-aware image regression datasets.

**Geo-Aware Image Classification.** The geo-aware image classification task aims at classifying a given image (e.g., species images, ground-level images, satellite images, etc.) into its correct category based on the image itself as well as its associated location metadata. Figure 1 illustrates how location encoders from TorchSpatial can be used to solve this task. Please refer to Appendix A.2 for a description of the model setup of the geo-aware image classification task. Based on our investigation, LocBench incorporates 7 geo-aware image classification datasets:

1. **BirdSnap:** An image dataset about bird species based on BirdSnap dataset [7] with location annotations by [42]. It consists of 19576 images of 500 bird species that are commonly found in North America. This dataset and the other two following are widely used by multiple studies[42, 46, 53] to demonstrate location encoder's capacity to significantly increase the fine-grained species classification accuracy.

2. **BirdSnap†:** An enriched BirdSnap dataset constructed by [42] by simulating locations, dates, and photographers from the eBrid dataset [70], containing 43470 images of 500 categories.

3. **NABirds†:** Another image dataset about North American bird species based on the NABirds dataset [72], the location metadata were also simulated from the eBrid dataset [70]. It contains 23699 images of 555 bird species categories.

4. **iNat2017:** The worldwide species recognition dataset used in the iNaturalist 2017 challenges [74] with 675170 images and 5089 unique categories. We add the location information retroactively provided by iNaturalist 2021[32]. Although its spatial distribution focuses on North America and Europe, it still covers the entire globe, which makes it one of the most spatially extensive and species-rich image dataset known to us.

5. **iNat2018:** The worldwide species recognition dataset used in the iNaturalist 2018 challenges [74] with 461939 images and 8142 unique categories. Although the original competition didn't provide coordinates, we add them to our benchmark as additional information from the same data source of iNaturalist 2021[32]. It has a similar spatial distribution with iNat2017, covering all continents. We choose these two datasets to evaluate location encoder's capacity to improve fine-grained species classification performance at the global level.

6. **YFCC:** YFCC100M-GEO100 dataset, an image dataset derived from Yahoo Flickr Creative Commons 100M dataset [79] and was annotated by [71], containing 88986 images over 100 everyday object categories with location annotations. Here, we denote this dataset as YFCC. YFCC is a comprehensive public dataset with images across the United States. Despite the relatively limited geographic coverage, we employ this dataset to measure location encoder's capacity for multifaceted image classification in addition to domain-specific image classification.

7. **fMoW:** Functional Map of the World dataset (denoted as fMoW) [15] is an RS image classification dataset, containing RS images with diverse land use types collected all over the world. It is composed of about 363K training and 53K validation remote sensing images which are classified into 62 different land use types. We use the fMoWrgb version of the fMoW dataset which are JPEG compressed version of these RS images with only the RGB bands.

**Geo-Aware Image Regression.** The geo-aware image classification task has a similar task setup as the classification task. The difference is the image target label is a continuous value that represents population density, forest coverage percentage, nightlights luminosity, and other indices at the given location. Figure 1 demonstrates how location encoders and image encoders can be used to solve these tasks. Please refer to Appendix A.3 for a description of the model setup. Based on our investigation, we select and preprocess 10 geo-aware image regression datasets based on MOSAIKS and SustainBench benchmarks [66, 84]:

1. **MOSAIKS population density:** This dataset uses daytime remote sensing images as covariables to predict population density at the corresponding locations. The observations were geographically sampled with the uniformly-at-random (UAR) strategy on the earth's surface. The MOSAIKS originally contains 100K population density records with coordinates, but less than half of them can be matched to remote sensing images on the dataset. We apply a log transformation of the labels and add 1 beforehand to avoid dropping zero-valued labels. After data cleaning, we get 425,637 observations uniformly distributed across the world.

2. **MOSAIKS forest cover:** According to [66], forest in this dataset is defined as vegetation greater than 5 meters in height, and measurements of forest cover are given at a raw resolution of roughly 30m by 30m. The estimation of forest cover rate was achieved by analysis of multiple spectral bands of remote sensing imagery, other than RGB bands used in this dataset. After similar data cleaning and preprocessing step, we get 498,106 observations at the global level.

3. **MOSAIKS nightlight luminosity:** Like forest cover rate, nightlight luminosity is also derived from satellite imagery, but not the RGB bands that most computer vision models work on, nor daytime remote sensing images we use as inputs in our benchmark. Specifically, luminosity in this dataset refers to the average radiance at night in 2015, provided by the Visible Infrared Imaging Radiometer Suite (VIIRS). Following the same data preprocess step, we offer 492,226 observations of nightlight luminosity with corresponding satellite images.

4. **MOSAIKS elevation:** Similarly, Satellite RGB bands are used to predict the elevation at the corresponding location. Following the same data preprocess step, we offer 498,115 elevation observations. To align with the settings of MOSAIKS, we did not apply a log transformation on elevation labels. The underlying data behind this dataset mainly comes from the Shuttle Radar Topography Mission (SRTM) at NASA's Jet Propulsion Laboratory (JPL), in addition to other open data projects.

5. **SustainBench Asset Index/Women BMI/Water Index/Child Mortality Rate/Sanitation Index/Women Education:** SustainBench is a set of benchmarks that aim to regress indices for the UN Sustainable Development Goals (SDGs) based on satellite images and street-level images. The remote sensing images are collected from diverse sources, including the Landsat 5/7/8, DMSP, and VIIRS satellites. The street-level images come from the platform Mapillary. The labels are originally derived from household-level survey data from the Demographic and Health Surveys (DHS) program and are aggregated as community-level. We did not use the original country-based splits and reset a train/test dataset split with an 80:20 ratio as the target of LocBench is to compare the effectiveness of location encoders across the globe.

For all those 10 datasets, we set a train/test dataset split with an 80:20 ratio and provide an option to resample the training dataset at any user-defined proportion for the convenience of users. Interestingly, locations have not been used as additional features for geo-aware image regression in the original MOSAIKS and SustainBench papers [66, 84]. Here, we will use these datasets to investigate the performance of various location encoders provided by our TorchSpatial model framework.

## 3.3 Evaluation Metrics

### 3.3.1 Overall Model Performance Evaluation Metrics

We first evaluate the overall performance of various location encoders on different LocBench datasets to align with existing benchmarks, such as iNaturalist[74] and MOSAIKS[66]. For geo-aware classification datasets, we use Top-1 accuracy, Top-3 accuracy, and Mean Reciprocal Rank (MRR) to measure the comprehensive performance of location encoders. For regression datasets, we utilize $R^2$, Mean Absolute Error (MAE), and Root Mean Squared Error (RMSE) to evaluate their performance.

### 3.3.2 Geographic Bias Evaluation Metrics

While there is no universally accepted definition of geographic bias, in this work, we interpret it as a subclass of model bias which refers to *a phenomenon in which an AI model performs differently across geographic regions and its predictions are biased toward some predominated regions* [39, 55]. The cause of geographic bias can be population differences, sampling bias, economic development differences, etc. Generally speaking, given two AI models with equal overall performances, we would prefer the model with lower geographic bias, which means *the possibility of encountering a wrong prediction is more uniform across the region of interest*. While there is increasing interest in studying geographic bias, most existing research focus on qualitative analysis of AI models' geographic bias [23, 56] or developing ad-hoc geographic bias measures for specific tasks [39] or models, e.g., large language models [55]. In this work, we propose a systematic and universally applicable geographic bias evaluation framework based on spatial autocorrelation, called **Geo-Bias Scores**[1].

Note that classic spatial autocorrelation (SA) measures [21] such as Moran's I [61] and Geary's C [27] cannot be used for geographic bias quantification, because these statistics are not numerically comparable across different spatial patterns, i.e., those generated from different models. Please refer to Appendix A.4.1 for a more detailed explanation. Therefore, we develop our bias metrics based on a newly proposed statistical measure called **spatial self-information (SSI)** [76], which is an information-theory-based generalization of the classic Moran's I statistics and ensures numerical

---

[1]The implementation of Geo-Bias Scores is available at: `https://github.com/seai-lab/PyGBS`

comparability across different spatial patterns, thus being suitable for geographic bias quantification. Please refer to Appendix A.4.2 for a detailed explanation of SSI.

Based on SSI we propose two novel metrics to quantify the geographic bias of model performance, which we call **unmarked SSI geo-bias score** and **marked SSI geo-bias score**, respectively. We assume that we evaluated our model on a test set with $M$ observations, and for each observation $i$ we get a performance measure $x_i$ (e.g., binary classification error, real-valued regression error, etc.) and criteria of high/low performance (e.g., wrong classification, larger than 3-sigma deviation). After applying the criteria to $x_i$, we get a set of binary values (e.g., -1 for low and 1 for high) of model performance. Together with the locations of these observations, we obtain a spatial sample on which we can compute the SSI. As many geospatial datasets are both large and distributed across the globe, it is important to consider the multi-scale effect of spatial patterns – some models may perform uniformly well on the continental scale, but are heavily biased towards mega-cities within continents, and vice versa. Thus, we further design our geo-bias evaluation metrics to be aware of spatial scales. We extract a neighborhood (e.g., within a 100km radius) for each low-performance observation $i$, draw a spatial grid as the background, construct the weight matrix by the spatial connectivity within this neighborhood (e.g., the nearest 4 locations are considered adjacent), and compute the SSI $J_i$ for this neighborhood.

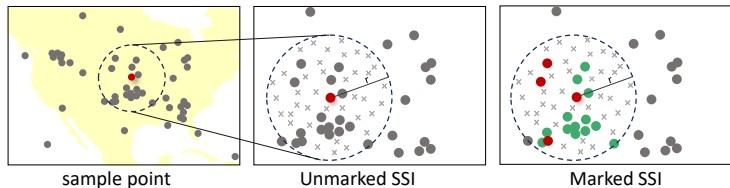

Figure 2: Intuition of the two geo-bias scores. **Left:** When we encounter a low-performance observation (red dot), we extract its neighborhood by radius $r$. **Middle:** Dots represent the observed locations and crosses are background grid points. Dots within the neighborhood demonstrate spatial patterns against the unobserved background. The SSI of such patterns is called the **unmarked SSI geo-bias score**. It reflects the intrinsic sampling geo-bias. **Right**: Green and red dots represent locations where the model achieves high performance and low performance respectively. The SSI of such patterns is called the **marked SSI geo-bias score**. It reflects the geo-bias of model performance, dependent both on where the data are observed and how the model performs at these locations.

There are two sources of SSI for a given neighborhood: (1) the spatial distribution of observations regardless of their model performances (referred to as *unmarked*, where we only consider the geographic locations of the data points without any additional attributes), and (2) the spatial distribution of the model performances (referred to as *marked*, where each data point is associated with an additional attribute, such as prediction accuracy or error). Intuitively, if a neighborhood itself is regularly arranged, no matter how random the low-performance observations scatter over the neighborhood, its SSI will still be high. Therefore, we call the SSI of source (1) the **unmarked SSI geo-bias score**. It measures the base strength of spatial patterns of the neighborhood we evaluate our model against. Starting from the unmarked SSI geo-bias, the spatial distribution of low-performance observations may further increase or decrease the SSI of the neighborhood. We define **marked SSI geo-bias score** as the difference between the SSI of the low-performance observations generated by our models and the SSI of completely random low-performance observations. This score measures the relative strength of spatial patterns of the low-performance observations. Please refer to Appendix A.4.3 for a detailed implementation procedure of geo-bias scores. In our experiments, we compute the unmarked SSI geo-bias scores and the marked SSI geo-bias scores for all neighborhoods that contain low-performance observations and report the average. Figure 2 illustrates the ideas of these two Geo-Bias scores.

It is intuitive to interpret the two scores. The unmarked SSI geo-bias score measures how likely low-performance observations occur in regions of strong spatial patterns. The marked SSI geo-bias score measures how likely low-performance observations themselves form strong spatial patterns. For both scores, *the larger the values, the stronger the spatial autocorrelation, indicating a higher geographic bias in the associated model.*

# 4 Experiments

In this section, we systematically evaluate the 15 location encoders developed in our TorchSpatial model framework on 7 geo-aware image classification and 10 regression datasets in LocBench. Both the overall model performance and Geo-Bias scores are reported.

## 4.1 Geo-Aware Image Classification

To test the effectiveness of 15 location encoders, we conduct experiments on 7 geo-aware image classification datasets including 5 fine-grained species recognition datasets, 1 Flickr image classification dataset, and 1 remote sensing image classification dataset as we described in Section 3.2. Besides 15 geo-aware classification models equipped with those 15 location encoders described in Section 3.1, we also consider *No Prior*, which represents a fully supervised trained image classifier without using any location information. In addition, we tested GPT-4V on the same datasets but in a zero-shot setting to see how it performs compared to fine-tuned geo-aware models. The Top-1 accuracy of all 17 models across 7 datasets is listed in Table 1, and the geo-bias scores are shown in Table 2.

Table 1: The Top1 classification accuracy of different models on 7 geo-aware image classification datasets in LocBench benchmark. See Appendix A.1 for the description of each model. We classify them into four groups: (A) No Prior indicates image-only models; (B) geo-aware models with 2D location encoders; (C) geo-aware models with 3D location encoders; (D) GPT-4V. Since the test sets for iNat2017, iNat2018, and fMoW are not open-sourced, we report results on their validation sets. The original result reported by [5] for No Prior on fMOW is 69.05. We obtain 69.83 by retraining their implementation. GPT-4V is tested with zero-shot settings, and * indicates that we resample 100 images from each dataset's test/validation set except BirdSnap and Birdsnap† whose whose test sets are used for evaluation. "Avg" column indicates the average performance of each model on all five species recognition datasets. **Bold** indicates the best models in Group B and C. See Section A.5 for hyperparameter tuning details.

| | Task | Species Recognition | | | | | | Flickr | RS |
|---|---|---|---|---|---|---|---|---|---|
| | Image Classification Dataset | BirdSnap | BirdSnap† | NABirds† | iNat2017 | iNat2018 | Avg | YFCC | fMOW |
| | $P(y\|\mathbf{x})$ - Prior Type | Test | Test | Test | Val | Val | - | Test | Val |
| A | No Prior (i.e. image model) | 70.07 | 70.07 | 76.08 | 63.27 | 60.20 | 67.94 | 50.15 | 69.83 |
| | *tile* [71] | 70.20 | 70.56 | 75.78 | 62.54 | 56.30 | 67.08 | 50.01 | 69.86 |
| | *wrap* [42] | **72.06** | 79.35 | **81.78** | 68.16 | 73.11 | 74.89 | 51.03 | 70.34 |
| | *wrap + ffn* [53] | 71.93 | 79.05 | 81.40 | **69.52** | 72.29 | 74.84 | 50.71 | 70.11 |
| B | *rbf*[48] | 71.79 | 79.58 | 81.74 | 68.24 | 70.03 | 74.28 | 51.22 | **70.68** |
| | *rff*[63] | 71.84 | 78.91 | 81.61 | 68.86 | 72.32 | 74.71 | 50.81 | 70.24 |
| | *Space2Vec-grid* [48] | 71.75 | **80.24** | 81.70 | 68.23 | 73.06 | 75.00 | **51.25** | 70.67 |
| | *Space2Vec-theory* [48] | 71.79 | 80.11 | 81.65 | 68.30 | **73.52** | **75.07** | 51.24 | 70.49 |
| | *xyz* [53] | 71.88 | 78.96 | 81.15 | 68.65 | 71.44 | 74.42 | 50.87 | 70.16 |
| | *NeRF [59]* | 72.10 | 79.93 | 81.62 | 68.74 | 72.91 | 75.06 | 51.27 | 70.60 |
| | *Sphere2Vec-sphereC* [53] | 72.10 | 79.97 | 81.91 | 69.34 | 72.93 | 75.25 | **51.35** | 70.85 |
| C | *Sphere2Vec-sphereC+* [53] | **72.15** | **80.90** | **82.13** | 68.29 | **73.45** | **75.38** | 51.31 | **70.93** |
| | *Sphere2Vec-sphereM*[53] | 71.88 | 79.93 | 81.86 | 68.51 | 72.94 | 75.02 | 51.18 | **70.93** |
| | *Sphere2Vec-sphereM+* [53] | 72.06 | 79.09 | 81.67 | 69.18 | 72.06 | 74.81 | 51.27 | 70.19 |
| | *Sphere2Vec-dfs* [53] | 71.79 | 78.69 | 81.44 | **69.42** | 72.16 | 74.70 | 50.65 | 70.27 |
| | *Siren (SH)*[67] | 71.88 | 78.96 | 81.72 | 67.68 | 71.33 | 74.29 | 50.57 | 70.20 |
| D | GPT-4V | 55.02 | 48.89 | 73.00* | 28.00* | 18.00* | 44.00* | 37.00* | 17.00* |

**Discussion.** According to Table 1, we can see that adding a location encoder can lead to significant model performance boosting. *Sphere2Vec-sphereC+* is the winner on 4 datasets except iNat2017, iNat2018, and YFCC in which *wrap + ffn*, *Space2Vec-theory*, and *Sphere2Vec-sphereC* are the winner respectively. Compared with other geo-aware models, GPT-4V demonstrates much worse performance. One probable reason is that fine-grained species recognition datasets usually contain hundreds or thousands of species classes which makes it hard for GPT-4V to handle. And RS images in fMoW are very different from natural images used to pre-train GPT-4V which leads to its poor performance. Further analysis of GPT-4V's performance on these geo-aware image classification tasks is needed. By comparing Table 1 and 2, we can see that except *tile*, all the other location encoders can significantly increase the model's geographic bias despite the overall model performance boosting. *tile* has relatively less impact on both overall model performance and geographic bias.

Table 2: Geo-bias scores of all location encoders across 7 geo-aware image classification datasets. `unmarked` represents the unmarked SSI geo-bias score, and `marked` represents the marked SSI geo-bias score. Both geo-bias scores are computed at the scale of 100km and using a 4-nearest-neighbor weight matrix. **Bold** numbers indicate that the scores that are significantly larger (>30%) than the `No Prior` model (i.e., the location-unaware model); * indicates the scores that are the largest among all models for this dataset. For GPT-4V, we do not report the geo-bias scores for larger datasets because our evaluation is limited to small subsets from these data (e.g., iNaturalist), due to budget constraints. Consequently, these geo-bias scores are not directly comparable to those of other studies.

| Task | Species Recognition | | | | | | | | | | | | Flickr | | RS | |
|---|---|---|---|---|---|---|---|---|---|---|---|---|---|---|---|---|
| Image Classification Dataset | BirdSnap | | BirdSnap† | | NABirds† | | iNat2017 | | iNat2018 | | | | YFCC | | fMOW | |
| $P(y\mid\mathbf{x})$ - Prior Type | Test | | Test | | Test | | Test | | Val | | | | Test | | Val | |
| Geo-Bias Score | unmarked | marked | unmarked | marked | unmarked | marked | unmarked | marked | unmarked | marked | unmarked | marked | unmarked | marked | unmarked | marked |
| A No Prior (i.e. image model) | 28.22 | 33.11 | 8.22 | 7.06 | 39.71 | 31.33 | 26.60 | 20.37 | 18.20 | 13.38 | | | 8.05 | 4.45 | 375.73 | 319.66 |
| B *tile* [71] | 27.65 | 32.10 | 8.53 | 7.37 | 38.43 | 30.26 | 26.08 | 19.91 | 16.80 | 12.22 | | | 8.41 | 4.77 | 375.43 | 319.77 |
| *wrap* [42] | 27.76 | 32.98 | **17.17** | **16.60** | **57.37** | **41.99** | **34.83** | **27.50** | **30.78** | **24.31** | | | 7.99 | 4.41 | 380.20 | 323.67 |
| *wrap* + *ffn* [53] | 29.50 | 34.99 | 8.25 | 7.07 | **57.03** | **42.43** | **35.73** | **28.20** | **27.68** | **21.57** | | | 7.77 | 4.21 | 377.41 | 321.20 |
| *rbf*[48] | 17.24 | 19.75 | 9.37 | 8.52 | **58.05** | **43.05** | **34.05** | **26.80** | 20.48 | 15.28 | | | 7.37 | 3.86 | 380.64 | 324.46 |
| *rff*[63] | 28.03 | 33.61 | **13.70** | **12.80** | **57.71** | **42.63** | **34.45** | **27.21** | **28.63** | **22.45** | | | 7.87 | 4.29 | 377.94 | 317.65 |
| *Space2Vec-grid* [48] | 22.26 | 25.10 | **16.27** | **15.42** | **58.96** | **43.38** | **34.10** | **26.87** | **31.12** | **24.71** | | | 7.99 | 4.43 | 380.23 | 323.17 |
| *Space2Vec-theory* [48] | **36.78*** | **42.98*** | 15.27 | 14.36 | **59.62** | **44.38*** | **34.12** | **26.87** | **31.68*** | **24.92** | | | 7.99 | 4.41 | 382.49 | 324.52 |
| C *xyz* [53] | 29.64 | 35.02 | **14.22** | **13.38** | **220.96*** | 34.09 | **34.89** | **27.53** | **26.33** | **20.44** | | | 7.79 | 4.24 | 379.84 | 323.12 |
| *NeRF* [59] | 29.66 | 35.16 | **16.13** | **15.53** | **57.86** | **42.61** | **34.93** | **27.62** | **30.46** | **23.90** | | | 7.81 | 4.26 | 375.81 | 320.30 |
| *Sphere2Vec-sphereC* [53] | 28.84 | 34.02 | **14.78** | **13.94** | **59.26** | **43.68** | **35.77*** | **28.21*** | **31.61** | **24.96*** | | | 7.67 | 4.16 | 377.07 | 320.78 |
| *Sphere2Vec-sphereC+* [53] | 30.43 | 36.48 | **19.99*** | **19.24*** | **59.13** | **43.47** | **33.14** | **26.02** | **31.55** | **24.85** | | | 8.22 | 4.66 | 379.92 | 323.04 |
| *Sphere2Vec-sphereM* [53] | 31.49 | 37.02 | **16.75** | **16.70** | **58.68** | **43.10** | **33.97** | **26.75** | **31.66** | **24.95** | | | 8.06 | 4.51 | 377.26 | 321.56 |
| *Sphere2Vec-sphereM+* [53] | 27.55 | 33.04 | **14.35** | **13.46** | **53.71** | **40.03** | **35.44** | **27.97** | **26.88** | **20.83** | | | 8.13 | 4.56 | 376.64 | 321.21 |
| *Sphere2Vec-dfs* [53] | 26.39 | 30.93 | **13.57** | **12.50** | **55.43** | **40.75** | **35.52** | **28.05** | **26.00** | **20.13** | | | 7.87 | 4.30 | 380.82 | 323.78 |
| *Siren (SH)*[67] | 27.67 | 32.91 | **14.87** | **14.50** | **57.57** | **42.60** | **35.47** | **28.07** | **26.24** | **20.26** | | | 7.68 | 4.15 | 377.23 | 321.15 |
| D GPT-4V | 28.58 | 34.01 | 7.06 | 6.21 | - | - | - | - | - | - | | | - | - | - | - |

**Qualitative analysis.** We employ hot spot analysis to measure and visualize the performance of geo-aware image classification models, which identify statistically significant spatial clusters of high values (hot spots) and low values (cold spots). We consider the HIT@1 (i.e., a binary measure indicating whether the predicted top class is equal to the label) of each observation as the measurement of performance and analysis of hot spots and cold spots. Hot spots of HIT@1 indicate that high-performance data points spatially cluster together while cold spots mean low-performance data points cluster together. Both of them indicate the geographic bias of model performance. Gray points are non-significant points that can not reject the null hypothesis, indicating a random spatial distribution of performance, thus no geographic bias. We generate the results of all models across 7 datasets and plot 9 of them in Figure 7 in Appendix A.9. It is clear that the geographic pattern of model performance is predominantly determined by the spatial distributions of datasets instead of models. Different models show subtle geo-performance differences on the same dataset, while one model can perform highly differently across multiple datasets.

## 4.2 Geo-Aware Image Regression

To demonstrate the generalizability of location encoders across tasks, we utilize all 15 location encoders on 10 geo-aware image regression tasks from LocBench. Table 3 and Table 4 compares the overall performance and geo-bias scores of different models on these tasks. We can see that similar to the image classification results shown in Table 1, adding location encoders can significantly boost model performance on population density, forest cover, elevation, and most other tasks. Location information does not show a significant impact on the nightlight luminosity prediction task. Meanwhile, Table 4 shows that adding location encoders in some cases will increase the model's geographic bias but not as much as we see in Table 2. This is mainly because the data points from the MOSAIKS datasets are uniformly-at-random (UAR) sampled from the globe.

## 5 Conclusion

In this work, we introduce TorchSpatial, a deep learning framework and benchmark to facilitate the development and evaluation of location encoders. A TorchSpatial model framework has been developed to support location encoder development. 15 widely used location encoders have been developed under TorchSpatial framework. A benchmark LocBench has been constructed which includes 7 geo-aware image classification datasets and 10 geo-aware image regression datasets. These datasets have been utilized to evaluate the performances of those 15 location encoders under different task setups, dataset sizes, and geographic coverage. Moreover, we introduce a systematic and universally applicable geographic bias evaluation framework called Geo-Bias Scores. Experiments show that adding a location encoder module to an existing computer vision model can significantly boost the model's overall performance. Meanwhile, this practice will aggravate the model's geographic bias

Table 3: The $R^2$ of different models on 10 geo-aware image regression datasets in LocBench benchmark. See Appendix A.1 for the description of each model. We classify them into three groups: (A) No Prior indicates image-only models; (B) geo-aware models with 2D location encoders; and (C) geo-aware models with 3D location encoders. Since these real-value target labels are difficult for GPT-4V to understand, we did not show the results of GPT-4V. **Bold** indicates the best models in Group B and C. See Section A.5 for hyperparameter tuning details.

| | Image Regression Dataset | Population Density | Forest Cover | Nightlight Luminosity | Elevation | Asset Index | Women BMI | Water Index | Child Mortality Rate | Sanitation Index | Women Edu |
|---|---|---|---|---|---|---|---|---|---|---|---|
| A | No Prior (i.e. image model) | 0.38 | 0.52 | 0.33 | 0.27 | 0.40 | 0.27 | 0.26 | 0.02 | 0.33 | 0.22 |
| B | *tile* [71] | 0.04 | 0.46 | 0.18 | 0.76 | 0.00 | 0.00 | 0.00 | 0.00 | 0.00 | 0.00 |
| | *wrap* [42] | 0.57 | 0.72 | 0.31 | **0.79** | 0.47 | 0.64 | 0.31 | 0.33 | 0.42 | 0.50 |
| | *wrap + ffn* [53] | 0.47 | 0.67 | 0.28 | 0.73 | 0.45 | 0.63 | 0.29 | 0.32 | 0.39 | 0.49 |
| | *rbf*[48] | 0.25 | 0.54 | **0.32** | 0.39 | 0.56 | **0.66** | 0.40 | **0.36** | 0.56 | 0.51 |
| | *rff*[63] | 0.57 | **0.73** | 0.23 | 0.77 | 0.50 | 0.64 | 0.33 | 0.34 | 0.46 | 0.53 |
| | *Space2Vec-grid* [48] | **0.65** | 0.69 | 0.22 | 0.76 | 0.66 | **0.66** | 0.49 | 0.32 | 0.59 | 0.64 |
| | *Space2Vec-theory* [48] | 0.57 | **0.73** | 0.21 | 0.78 | **0.70** | 0.65 | **0.52** | 0.33 | **0.61** | **0.66** |
| C | *xyz* [53] | 0.49 | 0.58 | 0.28 | 0.72 | 0.44 | 0.62 | 0.28 | 0.31 | 0.38 | 0.48 |
| | *NeRF [59]* | 0.60 | 0.68 | 0.23 | 0.76 | 0.65 | 0.68 | 0.50 | 0.34 | 0.60 | 0.64 |
| | *Sphere2Vec-sphereC* [53] | 0.63 | 0.73 | 0.28 | **0.82** | 0.69 | **0.69** | 0.52 | **0.37** | 0.62 | **0.66** |
| | *Sphere2Vec-sphereC+* [53] | **0.64** | **0.75** | 0.27 | **0.82** | 0.69 | 0.68 | **0.53** | **0.37** | **0.64** | **0.66** |
| | *Sphere2Vec-sphereM*[53] | 0.62 | 0.71 | 0.23 | **0.82** | 0.67 | 0.68 | 0.52 | **0.37** | 0.63 | **0.66** |
| | *Sphere2Vec-sphereM+* [53] | 0.53 | 0.67 | 0.32 | 0.74 | 0.45 | 0.62 | 0.29 | 0.31 | 0.39 | 0.48 |
| | *Sphere2Vec-dfs* [53] | 0.52 | 0.62 | **0.35** | 0.66 | 0.45 | 0.63 | 0.30 | 0.32 | 0.40 | 0.49 |
| | *Siren (SH)*[67] | 0.62 | 0.72 | 0.34 | 0.80 | 0.52 | 0.65 | 0.35 | 0.35 | 0.47 | 0.54 |

Table 4: Geo-bias scores of all location encoders across 4 geo-aware image regression datasets. `unmarked` represents the unmarked SSI geo-bias score, and `marked` represents the marked SSI geo-bias score. Both geo-bias scores are computed at the scale of 1000km and using a 4-nearest-neighbor weight matrix. **Bold** numbers indicate that the scores that are significantly larger (>30%) than the `No Prior` model (i.e., the location-unaware model); * indicates the scores that are the largest among all models for this dataset. Since we did not conduct regression modeling with GPT-4V, we did not show its results.

| | Image Regression Dataset | Population Density | | Forest Cover | | Nightlight Luminosity | | Elevation | |
|---|---|---|---|---|---|---|---|---|---|
| | Geo-Bias Score | unmarked | marked | unmarked | marked | unmarked | marked | unmarked | marked |
| A | No Prior (i.e. image model) | 5.93* | 2.53 | 6.71 | 2.73 | 7.47 | 0.71 | 6.71 | 2.92 |
| B | *tile* [71] | 5.40 | 2.34 | 5.73 | 2.38 | 7.60 | 0.54 | 6.62 | 2.87 |
| | *wrap* [42] | 4.86 | 2.01 | 5.17 | 2.27 | 7.36 | **1.08** | 5.71 | 2.74 |
| | *wrap + ffn* [53] | 5.04 | 1.90 | 5.55 | 2.40 | 7.61 | 0.27 | 6.12 | 2.90 |
| | *rbf*[48] | 5.39 | 2.02 | 5.83 | 2.28 | 7.69 | 0.42 | **7.28** | **3.37** |
| | *rff*[63] | 5.09 | 2.27 | 5.13 | 2.04 | 7.55 | 0.50 | 5.51 | 2.61 |
| | *Space2Vec-grid* [48] | 5.48 | 2.42 | 5.25 | 2.11 | 7.64 | 0.80 | 6.22 | 2.65 |
| | *Space2Vec-theory* [48] | 5.00 | 1.97 | 5.46 | 2.33 | 7.55 | 0.75 | 5.07 | 2.24 |
| C | *xyz* [53] | 5.64 | 2.38 | 5.65 | 2.37 | 7.55 | 0.54 | 5.96 | 2.97 |
| | *NeRF [59]* | 5.90 | 2.80* | 5.64 | 2.60 | 7.70 | 0.54 | 4.94 | 2.03 |
| | *Sphere2Vec-sphereC* [53] | 5.69 | 2.35 | 6.72* | 2.24 | 7.73* | 0.36 | 8.83* | 4.16* |
| | *Sphere2Vec-sphereC+* [53] | 5.34 | 2.37 | 5.08 | 2.18 | 7.50 | 0.74 | 5.49 | 2.51 |
| | *Sphere2Vec-sphereM* [53] | 5.21 | 2.39 | 5.12 | 2.31 | 7.58 | 0.78 | 5.01 | 2.21 |
| | *Sphere2Vec-sphereM+* [53] | 5.20 | 2.51 | 5.45 | 2.31 | 7.54 | 0.67 | 6.55 | 3.13 |
| | *Sphere2Vec-dfs* [53] | 5.45 | 2.59 | 6.08 | 2.94* | 7.53 | **1.10** | 6.03 | 2.58 |
| | *Siren (SH)*[67] | 5.10 | 2.29 | 5.82 | 2.46 | 7.48 | **1.21*** | 5.39 | 2.35 |

when the spatial distribution of original data samples is geographically biased. However, this practice will have little impact on the model's geographic bias if the original datasets are evenly sampled from the globe.

Our TorchSpatial currently only supports location encoder development. We plan to extend TorchSpatial's capability to support SRL for more spatial data types. Moreover, we plan to incorporate more geo-aware datasets and tasks into our LocBench such as sustainability index prediction [84], image geolocalization [30, 90], geographic question answering [44, 49, 20], weather forecasting, etc.

As far as we know, this is the first work that proposes a systematic and universally applicable geographic bias evaluation framework called Geo-Bias Scores. It can foster the responsible development of AI models, especially foundation models [9, 43], and ensure their spatial fairness.

## Acknowledgements

This work was supported by funds from the University of Georgia IIPA Seed Grand and UGA Presidential Interdisciplinary Seed Grant. Gengchen Mai acknowledges support from the Microsoft Research Accelerate Foundation Models Academic Research (AFMR) Initiative.

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

# A Appendix

## A.1 Fifteen Widely Used Location Encoders Implemented in TorchSpatial

We implement fifteen commonly recognized location encoders based on the TorchSpatial model framework. We classify them into two groups: 1) 2D location encoders which work on a projected 2D space [7, 2, 46, 63, 42, 46] and 2) 3D location encoders which interpret geolocation as 3D coordinates [47, 60, 67].

For location encoders applicable in a projected 2D space, TorchSpatial implements the following 7 models:

1. $tile$ is a discretization-based naive location encoder adopted by prior studies[7, 2, 71], diving space into partitions and then encoding locations with corresponding partition representations.
2. $wrap$ [42] is a sinusoidal location encoder, it uses a coordinate wrap mechanism to convert each dimension of location into 2 numbers and feed them into $\mathbf{NN}^{wrap}()$. $\mathbf{NN}^{wrap}()$ consists of four residual blocks which are implemented as linear layers.
3. $wrap + ffn$ [53] is a variant of $wrap$ that replaces $\mathbf{NN}^{wrap}()$ in $wrap$ with $\mathbf{NN}^{ffn}()$.
4. $rbf$ [46] is a kernel-based location encoder, it randomly samples $W$ points from the training dataset as RBF anchor points, and uses Gaussian kernels $\exp\left(-\dfrac{\parallel \mathbf{x}_i - \mathbf{x}_m^{anchor} \parallel^2}{2\sigma^2}\right)$ on each anchor points, where $\sigma$ is the kernel size. Each input point $\mathbf{x}_i$ is encoded as a $W$-dimension RBF feature vector, which is fed into $\mathbf{NN}^{ffn}()$ to obtain the location embedding.
5. $rff$ means *Random Fourier Features* [63, 62]. It first encodes location $\mathbf{x}$ into a $W$ dimension vector - $PE^{rff}(\mathbf{x}) = \varphi(\mathbf{x}) = \frac{\sqrt{2}}{\sqrt{W}} \bigcup_{i=1}^{W}[\cos{(\omega_i^T \mathbf{x} + b_i)}]$ where $\omega_i \overset{i.i.d}{\sim} \mathcal{N}(\mathbf{0}, \delta^2 I)$ is a direction vector whose each dimension is independently sampled from a normal distribution. $b_i$ is a shift value uniformly sampled from $[0, 2\pi]$ and $I$ is an identity matrix. Each component of $\varphi(\mathbf{x})$ first projects $\mathbf{x}$ into a random direction $\omega_i$ and makes a shift by $b_i$. Then it wraps this line onto a unit circle in $\mathbb{R}^2$ with the cosine function. $PE^{rff}(\mathbf{x})$ is further fed into $\mathbf{NN}^{ffn}()$ to get a location embedding.
6. *Space2Vec-grid* and *Space2Vec-theory* are two multi-scale location encoder on 2D Euclidean space proposed by [46]. Both of them implement the position encoder $PE^($ $\mathbf{x})$ as a deterministic Fourier mapping layer which is further fed into the $\mathbf{NN}^{ffn}()$. Both models' position encoders can be treated as performing a Fourier transformation on a 2D Euclidean space.

We also encompass 8 location encoders that learn location embeddings from 3D space as follows:

1. $xyz$ [53] first uses position encoder $PE^{xyz}(\mathbf{x})$ to convert the lat-lon spherical coordinates into 3D Cartesian coordinates centered at the sphere core. And then it feeds the 3D coordinates into a multilayer perceptron $\mathbf{NN}^{ffn}()$.
2. $NeRF$ can be treated as a multiscale version of $xyz$ using Neural Radiance Fields (NeRF) [60] for its position encoder.
3. *Sphere2Vec-sphereC*, *Sphere2Vec-sphereC+*, *Sphere2Vec-sphereM*, *Sphere2Vec-sphereM+* , *Sphere2Vec-dfs* are variants of *Sphere2Vec* [53], a multi-scale location encoder for spherical surface. They are the first location encoder series that preserves the spherical surface distance between any two points to our knowledge.
4. *Siren (SH)* [67] is another spherical location encoder proposed recently. It uses spherical harmonic basis functions as the position encoder $PE^{Siren\,(SH)}(\mathbf{x})$ and a sinusoidal representation network (SirenNets) as the $\mathbf{NN}()$.

## A.2 Geo-Aware Image Classification Model Setup

Figure 1 illustrates how location encoders from TorchSpatial can be used to solve this task. Following the common practice by [42, 46, 53, 51, 67], we use an image encoder and a location encoder to encode a given image $\mathbf{I}$ and its geolocation $\mathbf{x}$ respectively. Both encoders are trained separately in a supervised learning manner for species classification. In the inference stage, the probability of image category $y$ given the image and geolocation are multiplied together for final model prediction, i.e., $P(y|\mathbf{I}, \mathbf{x}) \propto P(y|\mathbf{I})P(y|\mathbf{x})$. Please refer to [42, 53] for a detailed explanation.

### A.3 Geo-Aware Image Regression Model Setup

As shown in Figure 1, a given image and its associated location are represented as an image embedding and a location embedding by an image encoder and a location encoder respectively. Then, we compute the Hadamard product (i.e., elementwise multiplication) of the image embedding and location embedding. The resulting embedding is fed into a multilayer perceptron (MLP) to regress the target image label.

### A.4 Geo-Bias Evaluation Framework

#### A.4.1 Why can't classic spatial autocorrelation statistics metrics be used for geographic bias quantification?

Spatial autocorrelation (SA) [28] refers to the phenomenon that the spatially distributed values occur not in a random manner. It is a high-dimensional generalization of the (temporal) autocorrelation in time series. A stronger spatial autocorrelation means that given a value at a location, the values of the nearby locations are more predictable, either more predictably similar (positive autocorrelation) or dissimilar (negative autocorrelation). In the situation of evaluating the geographic bias of model performances, strong spatial autocorrelation implies that the regions of consistently high/low performances form non-random spatial patterns (e.g., clusters). Traditionally, spatial autocorrelation statistics such as Moran's I and Geary's C are used to quantify whether spatial autocorrelation is statistically significant in a given region. However, these statistics are not numerically comparable across regions, i.e., one can not say if region A shows higher spatial autocorrelation than region B by comparing the values of their spatial autocorrelation statistics.

Assume that a spatial sample contains $N$ observations with values $\{x_i\}_{i=1}^N$. A weight matrix $W_{N \times N}$ is a non-negative real-valued matrix that represents the spatial connectivity between observations (e.g., the adjacency matrix based on $k$-nearest neighbors). The classic definition of the Moran's I is:

$$I = \frac{N}{\sum_{i=1}^N \sum_{j=1}^N W_{i,j}} \frac{\sum_{i=1}^N \sum_{j=1}^N W_{i,j}(x_i - \bar{x})(x_j - \bar{x})}{\sum_{i=1}^N (x_i - \bar{x})^2}$$

Statistically, if $I$ deviates largely from $-1/(N-1)$, we consider that there is significant spatial autocorrelation. However, it is easy to show from the definition that the values of classic Moran's I depend on the intrinsic variance of $x_i$, i.e., $\sum_{i=1}^N (x_i - \bar{x})^2$. When we compare the performance over two models, over two datasets, or over two regions, the intrinsic variances are different, and *the order of Moran's I values does not reflect the order of autocorrelation strength*. This causes severe problems if we wish to evaluate the geographic bias. For example, if we evaluate two geo-aware models – Model A and B – on the same geo-aware image classification/regression dataset, the evaluation results of them will form two spatial distribution patterns indicated as Pattern A and B. Each observation of these patterns indicates a model performance measure such as Top-1 accuracy for a classification task and absolute error for a regression task at a specific location. By computing SA metrics, e.g., Moran's I, we will get two measures of these two patterns. Individually, we can decide whether Pattern A or B shows spatial autocorrelation or not given its Moran's I score. However, we can not say Pattern A has a higher spatial autocorrelation than Pattern B by comparing their raw Moran's I scores. Thus, we cannot say Model A is more or less geographically biased than Model B.

#### A.4.2 A Brief Introduction of Spatial Self-Information

Different from Spatial autocorrelation (SA) [28], the spatial self-information [76] uses a Gaussian distribution to approximate the probability of observing certain types of spatial patterns. *The lower the probability, the less likely the current spatial patterns arise randomly, and consequently the stronger the spatial autocorrelation*. From an information-theoretic perspective of view, spatial self-information is the measure of the content of knowledge we can get by looking at the spatial arrangement of values: if a spatial sample has very low spatial self-information, it means the values are completely randomly spatially arranged, and knowing the spatial locations will not provide any useful information for predicting the values of the sample. On the contrary, if a spatial sample has very high self-information, it means knowing the spatial locations will increase our certainty in predicting the values of the sample. In evaluating geographic bias, if the model performances show high spatial self-information, it can be interpreted as the distribution of high/low performance regions

provides extra information than random, uniform distributions, i.e., the model systematically biased for/against certain locations.

Intuitively, SSI computes the negative log probability of observing a certain spatial pattern of data under the hypothesis that the data appear in space completely randomly. The larger the value, the more non-trivial the pattern is, indicating that the data are spatially biased. Briefly speaking, [76] proved that such a probability distribution is approximately Gaussian, with mean $\mu = \sum_{p \neq q}(c_p - \bar{x})(c_q - \bar{x})\mu_{p,q} + \sum_p (c_p - \bar{x})^2 \mu_{p,p}$ and variance $\sigma^2 = \sum_{p \neq q \neq r_{max}} \left[(c_p - \bar{x})(c_q - \bar{x}) - 2(c_p - \bar{x})(c_{r_{max}} - \bar{x}) + (c_{r_{max}} - \bar{x})^2\right]\sigma_{p,q}^2 + \sum_{p \neq r_{max}} \left[(c_p - c_{r_{max}})^2\right]\sigma_{p,p}^2$.

For detailed explanations on how to compute these terms, please refer to [76], Section 3.5, Theorem 8, and Theorem 9 for full mathematical formulas.

### A.4.3 Implementation Procedure of Geo-bias Scores

Suppose we have a set of locations $\{l_1, l_2, \ldots l_n\}$ and a set of corresponding binary ("High" or "Low") performance values $\{p_1, p_2, \ldots p_n\}$. Choose hyperparameter $k$, the maximum number of observations in the neighborhood considered in the geo-bias score computation; hyperparameter $r$, the maximum radius of neighborhood considered in the geo-bias score computation; and hyperparameter $d$, the density of background grid points.

Implementation Procedure:

1. For each $l_i$ **that is low-performance**, find its nearest $k$ neighbors $\{l_{i1}, l_{i2}, \ldots l_{ik}\}$ that fall into the circle of radius $r$, called neighborhood.

2. Construct background grid points $\{b_1, b_2, \ldots b_B\}$ within the neighborhood, satisfying that $B/\pi r^2 = d$. Set their values to be all 0s.

3. Compute the unmarked SSI geo-bias score: set the performance values of $\{l_{i1}, l_{i2}, \ldots l_{ik}\}$ to be all 1s, mix up with the background points, and compute the average spatial self-information (SSI) for this set of points.

4. Compute the marked SSI geo-bias score: set the performance values of $\{l_{i1}, l_{i2}, \ldots l_{ik}\}$ to be 1 for high performance and -1 for low performance. Mix up with the background grid points, and compute the average spatial self-information (SSI) for this set of points.

The setting of hyperparameters $(k, r)$ can be based on prior knowledge, i.e., domain knowledge that mandates certain choice of hyperparameters. For example, in ecology, $k$ may refer to the number of observations of wild lives and $r$ may refer to the migration range. We can get the best choice of $k$ and $r$ based on the experience of the ecologists. In case the users do not have domain knowledge for hyperparameter setting, we have the following rule of thumb (not necessarily optimal but statistically stable) strategy: Given the dataset, we select a $k$ larger than 30 to ensure statistical significance. We recommend around 100. We compute the k-nearest neighbors of each data point. Then, we compute the average distance of the farthest neighbor and use it as $r$. This is also the way how we selected the hyperparameters for our reported experiments and it proves working fairly well. As for $d$, we recommend setting it to be around 4 to 8 times larger than the observation density of the dataset.

### A.5 Model Tuning

TorchSpatial are implemented in PyTorch framework and all our experiments were conducted on a Ubuntu workstation equipped with 4 NVIDIA RTX A5500 GPUs each of which has 24 GB memory.

To find the best hyperparameter combinations for each model on each dataset, we use grid search to get rough ranges of each hypereparameter as the first step. Then we employ optuna [3], a hyperparameter optimization framework, to obtain the optimized hyperparameters. The hyperparameters included in the process of tuning are as follows: supervised learning rate $lr$, number of scales $S$, minimum scaling factor $r_{\min}$, number of hidden layers $h$ and number of neurons $k$ used in $\mathbf{NN}^{ffn}(\cdot)$. For $rbf$, the number of kernels $M$ and kernel size $\sigma$ are also considered during tuning. For $rff$, we also include kernel size. We also test various activation functions, such as Relu, LeakyRelu, and Sigmoid. We do not tune the $r_{\max}$, which is set as 1 in most cases but 360 for *Space2Vec-grid*.

The importance of each hyperparameter can be very diverse across different models and datasets. In general, the learning rate $lr$ is the most influential hyperparameter, $r_{\min}$ in *spacevec* and *Sphere2Vec* models is less important, and $h$ is the least influential one. Our practice proves that one hidden layer works best for most location encoders which is align with the findings in *Sphere2Vec* [53].

All pre-trained models and corresponding settings, as well as hyperparameter analysis such as contour map and bar chart, can be found on our TorchSpatial GitHub repository `https://github.com/seai-lab/TorchSpatial`.

## A.6 More Experiment Results

Table 5: The MRR of different models on 7 geo-aware image classification datasets in LocBench benchmark. See Appendix A.1 for the description of each model. We classify them into three groups: (A) No Prior indicates image-only models; (B) geo-aware models with 2D location encoders; (C) geo-aware models with 3D location encoders. Since the test sets for iNat2017, iNat2018, and fMoW are not open-sourced, we report results on their validation sets. "Avg" column indicates the average performance of each model on all five species recognition datasets. **Bold** indicates the best models in Group B and C. See Section A.5 for hyperparameter tuning details.

| | Task | Species Recognition | | | | | | Flickr | RS |
|---|---|---|---|---|---|---|---|---|---|
| | Image Classification Dataset | BirdSnap | BirdSnap† | NABirds† | iNat2017 | iNat2018 | Avg | YFCC | fMOW |
| | $P(y\|\mathbf{x})$ - Prior Type | Test | Test | Test | Val | Val | - | Test | Val |
| A | No Prior (i.e. image model) | 0.790 | 0.790 | 0.841 | 0.728 | 0.705 | 0.771 | 0.644 | 0.785 |
| B | *tile* [71] | 0.790 | 0.787 | 0.839 | 0.727 | 0.673 | 0.763 | 0.642 | 0.785 |
| | *wrap* [42] | **0.801** | 0.856 | **0.881** | 0.764 | 0.807 | 0.822 | 0.651 | 0.790 |
| | *wrap + ffn* [53] | **0.801** | 0.856 | 0.877 | **0.778** | 0.804 | 0.823 | 0.650 | 0.788 |
| | *rbf* [48] | 0.786 | 0.861 | 0.880 | 0.767 | 0.735 | 0.806 | 0.654 | **0.793** |
| | *rff* [63] | 0.799 | 0.852 | 0.880 | 0.767 | 0.800 | 0.820 | 0.651 | 0.791 |
| | *Space2Vec-grid* [48] | 0.789 | 0.860 | 0.880 | 0.767 | 0.807 | 0.820 | **0.656** | **0.793** |
| | *Space2Vec-theory* [48] | 0.798 | **0.862** | 0.880 | 0.767 | **0.812** | **0.824** | 0.655 | 0.791 |
| C | *xyz* [53] | 0.801 | 0.854 | 0.875 | 0.768 | 0.796 | 0.819 | 0.651 | 0.788 |
| | *NeRF* [59] | **0.802** | 0.862 | 0.880 | 0.771 | **0.809** | 0.825 | 0.649 | 0.786 |
| | *Sphere2Vec-sphereC* [53] | **0.802** | **0.863** | **0.881** | **0.777** | 0.806 | **0.826** | **0.657** | 0.793 |
| | *Sphere2Vec-sphereC+* [53] | 0.801 | **0.863** | 0.877 | 0.757 | **0.809** | 0.821 | 0.656 | 0.792 |
| | *Sphere2Vec-sphereM* [53] | 0.801 | 0.861 | **0.881** | 0.757 | 0.806 | 0.821 | 0.655 | **0.795** |
| | *Sphere2Vec-sphereM+* [53] | 0.800 | 0.854 | 0.875 | 0.775 | 0.803 | 0.821 | 0.642 | 0.785 |
| | *Sphere2Vec-dfs* [53] | 0.800 | 0.852 | 0.879 | 0.776 | 0.801 | 0.822 | 0.648 | 0.789 |
| | *Siren (SH)* [67] | 0.800 | 0.854 | 0.880 | **0.777** | 0.796 | 0.821 | 0.648 | 0.789 |

## A.7 Ethics of LocBench

We are using publically available data to construct our LocBench benchmark. There is no personally identifiable information included in our LocBench.

## A.8 The Spatial Distributions of different geo-aware dataset in LocBench

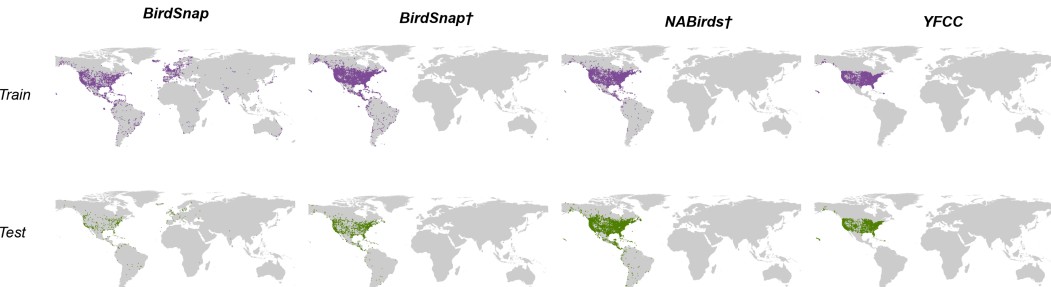

Figure 3: The spatial distributions of 4 geo-aware image classification datasets: BirdSnap, BirdSnap†, NABird†, and YFCC.

Table 6: The RMSE of different models on 4 geo-aware image regression datasets in LocBench benchmark. See Appendix A.1 for the description of each model. We classify them into three groups: (A) No Prior indicates image-only models; (B) geo-aware models with 2D location encoders; and (C) geo-aware models with 3D location encoders. Since these real-value target labels are difficult for GPT-4V to understand, we did not show the results of GPT-4V. **Bold** indicates the best models in Group B and C. See Section A.5 for hyperparameter tuning details.

| | Image Regression Dataset | Population Density | Forest Cover | Nightlight Luminosity | Elevation |
|---|---|---|---|---|---|
| A | No Prior (i.e. image model) | 1.833 | 1.270 | 0.338 | 790.057 |
| B | *tile* [71] | 1.828 | 1.361 | 0.345 | 1053.870 |
| | *wrap* [42] | **1.206** | **1.001** | 0.298 | 391.206 |
| | *wrap + ffn* [53] | 1.461 | 1.113 | 0.312 | 435.678 |
| | *rbf*[48] | 1.622 | 1.281 | 0.299 | 703.678 |
| | *rff*[63] | 1.407 | 1.007 | 0.316 | 407.502 |
| | *Space2Vec-grid* [48] | 1.372 | 1.015 | 0.338 | 398.446 |
| | *Space2Vec-theory* [48] | 1.598 | 1.060 | **0.277** | **390.352** |
| C | *xyz* [53] | 1.766 | 1.183 | 0.348 | 466.528 |
| | *NeRF [59]* | 1.647 | 1.084 | 0.324 | 413.007 |
| | *Sphere2Vec-sphereC* [53] | 1.471 | **0.967** | 0.314 | 371.175 |
| | *Sphere2Vec-sphereC+* [53] | 1.525 | 0.986 | 0.315 | **343.209** |
| | *Sphere2Vec-sphereM*[53] | 1.150 | 1.009 | 0.327 | 351.544 |
| | *Sphere2Vec-sphereM+* [53] | 1.282 | 1.092 | 0.289 | 432.246 |
| | *Sphere2Vec-dfs* [53] | 1.325 | 1.074 | 0.338 | 488.589 |
| | *Siren (SH)*[67] | **1.114** | 0.970 | **0.288** | 375.042 |

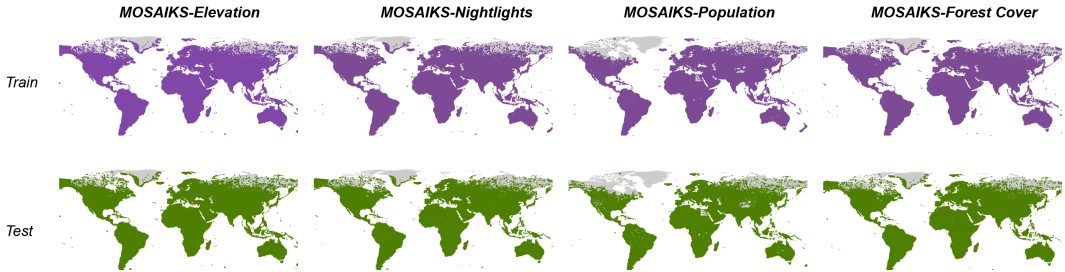

Figure 4: The spatial distributions of 3 geo-aware image classification datasets: iNat2017, iNat2018, and fMoW.

Figure 5: The spatial distributions of 4 geo-aware image regression datasets: MOSAIKS series.

## A.9 The Hot Spot Analysis of Three Geo-Aware Image Classification Datasets

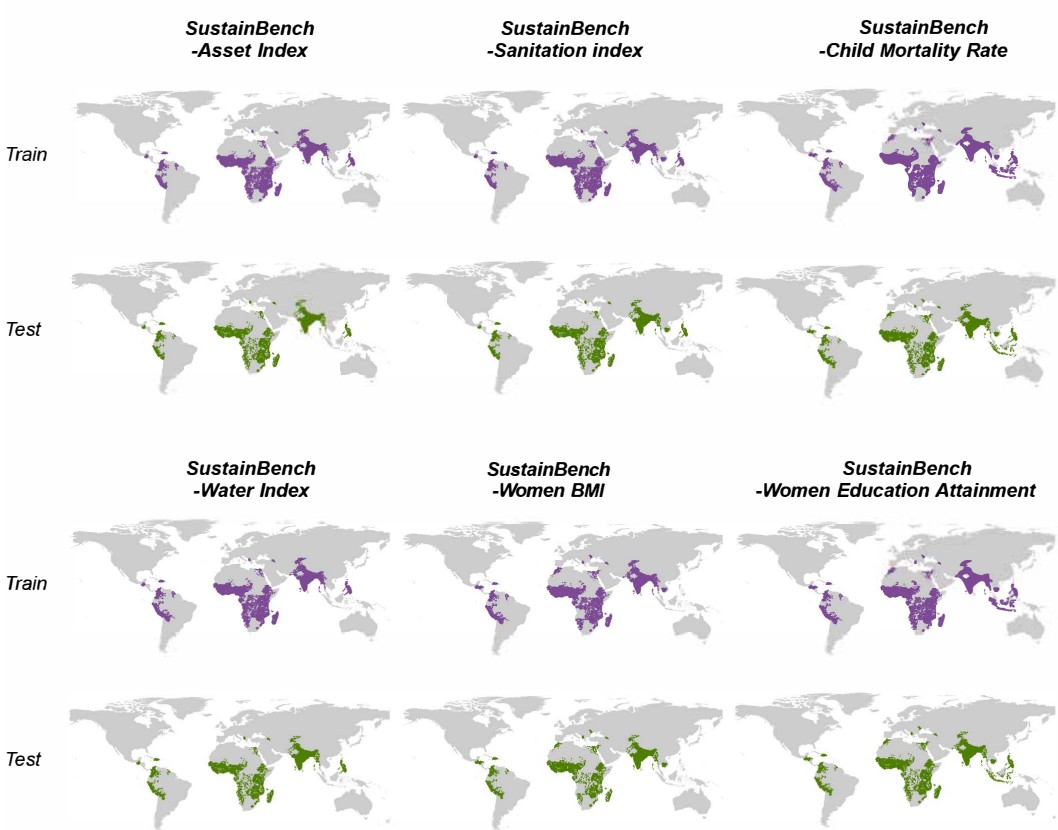

Figure 6: The spatial distributions of 6 geo-aware image regression datasets: SustainBench series.

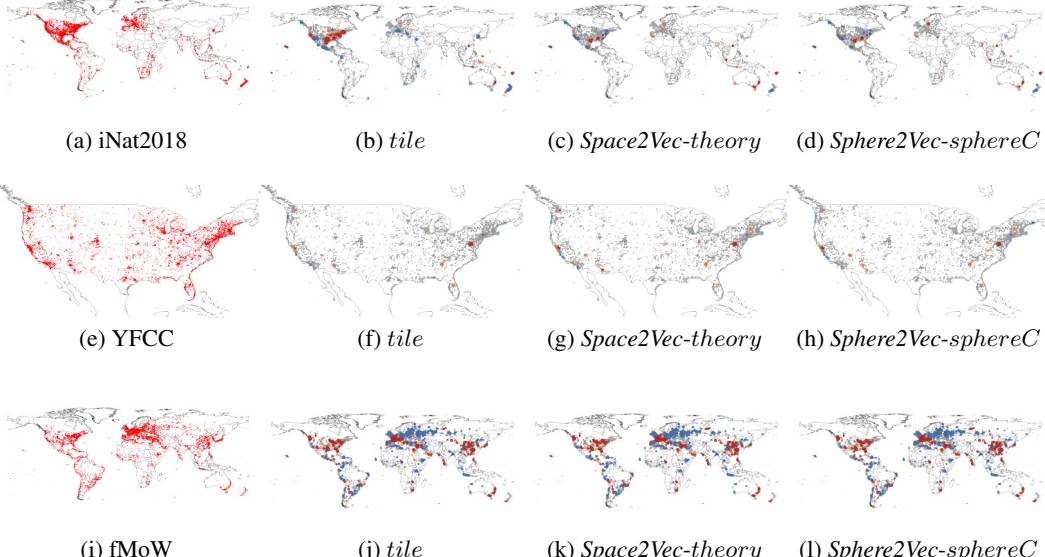

(a) iNat2018    (b) *tile*    (c) *Space2Vec-theory*    (d) *Sphere2Vec-sphereC*

(e) YFCC    (f) *tile*    (g) *Space2Vec-theory*    (h) *Sphere2Vec-sphereC*

(i) fMoW    (j) *tile*    (k) *Space2Vec-theory*    (l) *Sphere2Vec-sphereC*

Figure 7: Hot spot analysis of HIT@1 of three models on three datasets. The first column presents the spatial distributions of the test/validation datasets of iNat2018, YFCC, and fMoW. For the other three columns, red dots indicate hot spots, namely location clusters, where models tend to get good performance. In contrast, models generally perform poorly in blue areas. Gray indicates no significant spatial autocorrelation pattern found.

