# Supplementary Material: TorchSpatial-A Location Encoding Framework and Benchmark for Spatial Representation Learning

**Nemin Wu**[1*], **Qian Cao**[1*], **Zhangyu Wang**[2], **Zeping Liu**[3], **Yanlin Qi**[4],
**Jielu Zhang**[1], **Joshua Ni**[5], **Xiaobai Yao**[1], **Hongxu Ma**[6], **Lan Mu**[1],
**Stefano Ermon**[7], **Tanuja Ganu**[8], **Akshay Nambi**[8], **Ni Lao**[6†], **Gengchen Mai**[3,1†]

[1]University of Georgia,   [2]UC Santa Barbara,   [3]University of Texas at Austin,
[4]UC Davis,   [5]Basis Independent Fremont,   [6]Google LLC,
[7]Stanford University,   [8]Microsoft Research,
{nemin.wu, qian.cao1, jielu.zhang, xyao, mulan}@uga.edu,
zhangyuwang@ucsb.edu,   zeping.liu@utexas.edu,   ylqi@ucdavis.edu,
nijoshua2025@gmail.com,   {hxma, nlao}@google.com, ermon@cs.stanford.edu,
{tanuja.ganu, akshay.nambi}@microsoft.com,   gengchen.mai@austin.utexas.edu
*Equal contribution. Author ordering is determined by coin flip.   †Corresponding author.

## 1   Motivation

**For what purpose was the dataset created? Was there a specific task in mind?** Was there a specific gap that needed to be filled? Please provide a description.

In order to systematically compare the location encoders' performance and their impact on the model's overall geographic bias on datasets with various set setups, sizes, and geographic coverages, we clean and preprocess 7 geo-aware image classification datasets and 10 geo-aware image regression datasets.

**Geo-Aware Image Classification.**   The geo-aware image classification task aims to classify a given image (e.g., species images, ground-level images, satellite images, etc.) into its correct category based on the image itself and its associated location metadata. Figure 1 in our paper illustrates how location encoders from TorchSpatial can be used to solve this task. Please refer to Appendix Section 2 for a description of the model setup of the geo-aware image classification task.

**Geo-Aware Image Regression.**   The geo-aware image regression task has a similar task setup as the classification task. The difference is the image target label is a continuous value that represents population density, forest coverage percentage, elevation,nightlight luminosity, asset wealth index, child mortality rate, women BMI, women educational attainment, clean water index, and sanitation index at the given location.

**Who created the dataset (e.g., which team, research group) and on behalf of which entity (e.g., company, institution, organization)?**

Spatially Explicit Artificial Intelligence Lab from the University of Georgia & University of Texas at Austin created the dataset.

**Who funded the creation of the dataset?** If there is an associated grant, please provide the name of the grantor and the grant name and number.

This work was supported by funds from the University of Georgia IIPA Seed Grand and UGA Presidential Interdisciplinary Seed Grant. Dr. Gengchen Mai acknowledges the Microsoft Research Accelerate Foundation Models Academic Research (AFMR) Initiative for their support.

Submitted to 38th Conference on Neural Information Processing Systems (NeurIPS 2024). Do not distribute.

27  **Any other comments?**

28  None.

## 2   Composition

30  **What do the instances that comprise the dataset represent (e.g., documents, photos, people,**
31  **countries)?**  Are there multiple types of instances (e.g., movies, users, and ratings; people and
32  interactions between them; nodes and edges)? Please provide a description.

33  The instances in all 17 datasets represent images. We provide pretrained image embeddings as well
34  as the associated geolocations for all tasks. For the image classification tasks, the labels are the class
35  IDs, while for the regression tasks, the labels represent the predicted regression values.

36  **How many instances are there in total (of each type, if appropriate)?**

37  The number of instances for each dataset is listed in Table 1

Table 1: Dataset Information

| Task Category | Dataset | Instances |
|---|---|---|
| Image Classification | BirdSnap | 19576 |
| | BirdSnap† | 43470 |
| | NABirds† | 23699 |
| | iNat2017 | 675170 |
| | iNat2018 | 461939 |
| | YFCC | 36146 |
| | fMoW | 416612 |
| Image Regression | Population Density | 425637 |
| | Forest Cover | 498106 |
| | Nightlight Luminosity | 492226 |
| | Elevation | 498115 |
| | Asset Index | 89936 |
| | Women BMI | 94866 |
| | Water Index | 86938 |
| | Child Mortality Rate | 105582 |
| | Sanitation Index | 89271 |
| | Women Edu | 117062 |

38  **Does the dataset contain all possible instances or is it a sample (not necessarily random)**
39  **of instances from a larger set?**  If the dataset is a sample, then what is the larger set? Is the
40  sample representative of the larger set (e.g., geographic coverage)? If so, please describe how
41  this representativeness was validated/verified. If it is not representative of the larger set, please
42  describe why not (e.g., to cover a more diverse range of instances, because instances were withheld
43  or unavailable).

44  The 7 datasets for image classification and 6 datasets from SustainBench are not samples of instances
45  from a larger set but represent the whole datasets. We performed preprocessing and cleaning on
46  the four datasets for image regression from the dataset MOSAIKS. The process can be seen in the
47  "Collection Process" below. We plot the geographic distribution of these four datasets (see Appendix
48  Figure 5, and it shows that they are uniformly distributed, just like the original MOSAIKS.

49  **What data does each instance consist of?**  "Raw" data (e.g., unprocessed text or images)or features?
50  In either case, please provide a description.

51  We provide image features with geographic coordinates and labels (species, object categories, land
52  use types, population density, forest cover ratio, nightlight luminosity, elevation, asset wealth index,
53  child mortality rate, women BMI, women educational attainment, clean water index, and sanitation
54  index).

55  **Is there a label or target associated with each instance?**  If so, please provide a description.

The labels represent species, object categories, and land use types for classification tasks. For regression tasks, the labels are continuous values representing population density, forest cover ratio, nightlight luminosity, elevation, asset wealth index, child mortality rate, women BMI, women educational attainment, clean water index, and sanitation index as described above.

**Is any information missing from individual instances?** If so, please provide a description, explaining why this information is missing (e.g., because it was unavailable). This does not include intentionally removed information, but might include, e.g., redacted text.

Some coordinates are missing in the **BirdSnap**, **BirdSnap†**, **NABirds†**, **iNat2017**, and **iNat2018** datasets, and these coordinates are not available in their original sources. Despite this, we did not exclude these instances when evaluating the location encoders. This decision was made to fairly assess the performance of the encoders in real-world scenarios, where spatial information may be incomplete or unavailable sometimes. Including these instances in the evaluation can help evaluate the robustness and generalizability of the location encoders, ensuring they can effectively handle situations where not all instances are guaranteed to have complete spatial information.

**Are relationships between individual instances made explicit (e.g., users' movie ratings, social network links)?** If so, please describe how these relationships are made explicit.

Instances have geographic relationships with each other, which can be reflected in their coordinates. Otherwise, there are no explicit relations among different instances.

**Are there recommended data splits (e.g., training, development/validation, testing)?** If so, please provide a description of these splits, explaining the rationale behind them.

The datasets for image classification were already divided by their original sources. For the regression tasks, we performed an 8:2 random split on the datasets to form the training and testing datasets.

**Are there any errors, sources of noise, or redundancies in the dataset?** If so, please provide a description.

The geolocations of each species' images in **BirdSnap**, **BirdSnap†**, **NABirds†**, **iNat2017**, **iNat2018**, and **SustainBench** datasets might contain noise. The location metadata of each image includes a location uncertainty measure. Coordinates in **SustainBench** may have been randomly "offset" by up to 2 km in urban areas and 10 km in rural areas to safeguard the privacy of survey participants.

**Is the dataset self-contained, or does it link to or otherwise rely on external resources (e.g., websites, tweets, other datasets)?** If it links to or relies on external resources, a) are there guarantees that they will exist, and remain constant, over time; b) are there official archival versions of the complete dataset (i.e., including the external resources as they existed at the time the dataset was created); c) are there any restrictions (e.g., licenses, fees) associated with any of the external resources that might apply to a dataset consumer? Please provide descriptions of all external resources and any restrictions associated with them, as well as links or other access points, as appropriate.

The dataset is entirely self-contained.

**Does the dataset contain data that might be considered confidential (e.g., data that is protected by legal privilege or by doctor–patient confidentiality, data that includes the content of individuals' nonpublic communications)?** If so, please provide a description.

The 17 datasets contained in LocBench are open-sourced datasets that do not contain confidential information.

**Does the dataset contain data that, if viewed directly, might be offensive, insulting, threatening, or might otherwise cause anxiety?** If so, please describe why.

No, the dataset does not contain data that, if viewed directly, might be offensive, insulting, threatening, or might otherwise cause anxiety. The dataset is composed of images and labels related to natural elements and general socio-economic information. It does not include content that could be considered sensitive or inappropriate.

**Does the dataset identify any subpopulations (e.g., by age, gender)?** If so, please describe how these subpopulations are identified and provide a description of their respective distributions within the dataset.

No, the dataset is homogeneous in terms of these demographic factors, and all instances are considered as part of a single population without further differentiation.

**Is it possible to identify individuals (i.e., one or more natural persons), either directly or indirectly (i.e., in combination with other data) from the dataset?** If so, please describe how.

In the **BirdSnap**, **BirdSnap†**, **NABirds†**, **iNat2017**, and **iNat2018** datasets, the image metadata contains "user ID" which indicates who took the corresponding photo. While previous work [6] uses the user ID to generate user embeddings aiming at further improving the model performance, we do not consider them in the TorchSpatial since the focus of TorchSpatial is supporting the development and evaluation of location encoders.

**Does the dataset contain data that might be considered sensitive in any way (e.g., data that reveals race or ethnic origins, sexual orientations, religious beliefs, political opinions or union memberships, or locations; financial or health data; biometric or genetic data; forms of government identification, such as social security numbers; criminal history)? If so, please provide a description.**

No. There are no personal sensitive data contained in LocBench.

**Any other comments?**

None.

# 3   Collection Process

**How was the data associated with each instance acquired?** Was the data directly observable (e.g., raw text, movie ratings), reported by subjects (e.g., survey responses), or indirectly inferred/derived from other data (e.g., part-of-speech tags, model-based guesses for age or language)? If the data was reported by subjects or indirectly inferred/derived from other data, was the data validated/verified? If so, please describe how.

**BirdSnap**, **BirdSnap†**, and **NABirds†** were geographically annotated by [6] based on location simulation from the eBrid dataset[12], while their species labels are from the original BirdSnap[1] and NABirds[14]. For **iNat2017** and **iNat2018**, their species labels are also from the original iNaturalist 2017[15] and iNaturalist 2018 challenges [15], but the location annotations were provided by iNaturalist 2021[5]. **YFCC** was manually verified and annotated by [13] from a large set of the Yahoo Flickr Creative Commons 100M dataset [16], which was initially noisily tagged by Flickr users. **fMoW** was annotated by 642 GeoHIVE users according to [2]. In terms of four datasets for image regression from the **MOSAIKS**. Tree cover ratio, elevation, and nightlight luminosity were observed by satellite and estimated by researchers[4, 3, 10], while population density is from Gridded Population of the World (GPW) dataset v4[1]. For six **SustainBench** datasets, labels were derived from the Demographic and Health Surveys (DHS), which were originally household-level and were aggregated to community-level by [17].

**What mechanisms or procedures were used to collect the data (e.g., hardware apparatuses or sensors, manual human curation, software programs, software APIs)?** How were these mechanisms or procedures validated?

The species recognition datasets including **BirdSnap**, **BirdSnap†**, **NABirds†**, **iNat2017**, and **iNat2018** datasets are collected based on citizen science platforms. The **YFCC** dataset was constructed by using Flickr API. The **fMoW** dataset was originally constructed by [2] based on multiple satellite sensors and annotated by 642 GeoHIVE users. The four **MOSAIKS** datasets were originally collected by [11] based on collocated satellite images from Google Static Maps API, nightlight images from the Visible Infrared Imaging Radiometer Suite (VIIRS)[2], population data from GPW v4, tree cover data from [4], and elevation provided by Mapzen and accessed via the Amazon Web Services (AWS) Terrain Tile service. The six **SustainBench** datasets were originally collected by [17], whose images were from Landsat 5/7/8 satellites, the DMSP and VIIRS satellites, and labels were from the Demographic and Health Surveys (DHS) program.

---

[1]These data can be accessed at `https://sedac.ciesin.columbia.edu/data/collection/gpw-v4`.

[2]These data can be accessed at `https://eogdata.mines.edu/products/vnl/`.

**If the dataset is a sample from a larger set, what was the sampling strategy (e.g., deterministic, probabilistic with specific sampling probabilities)?**

All datasets included in the paper are not samples from larger sets.

**Who was involved in the data collection process (e.g., students, crowdworkers, contractors) and how were they compensated (e.g., how much were crowdworkers paid)?**

LocBench are constructed by preprocessing and cleaning multiple existing datasets by authors. There are no other people involved in this process.

**Over what timeframe was the data collected?** Does this timeframe match the creation timeframe of the data associated with the instances (e.g., recent crawl of old news articles)? If not, please describe the timeframe in which the data associated with the instances was created.

**BirdSnap** and **BirdSnap†** are derived from the BirdSnap, a dataset of bird images collected from the internet. The exact timeframe for data collection is not specified, but we can reasonably assume they were collected before the dataset's release year in 2014. **NABirds†**'s train locations and images are sampled from eBird 2015, and the test set is from 2016. **iNat2017** and **iNat2018** are part of the iNaturalist project, which crowdsources observations of biodiversity. The data in **iNat2017** and **iNat2018** were collected, annotated, cleaned, and released during the respective years (2017 and 2018). **YFCC** is a subset of YFCC100M, which contains images from Flickr in 2004 until early 2014. The location annotation was conducted by [13], so it was annotated before the publishing year 2015. **fMoW** contains remote sensing images from 2002 to 2017 and the distribution over the years is illustrated in [2]. The dataset was released in 2018. The **Forest Cover** is measured from data in 2010. The **Population Density** is originally from the GPW v4, which collected data between 2005 and 2014. In addition, population density in the US is from the 2010 census. **Nightlight Luminosity** derives from the 2015 annual composite of VIIRS. **Elevation** is composed of data roughly from 2010 to 2017. **SustainBench** contains nighttime remote sensing images from two sources: DMSP taken in 2011 or earlier, and VIIRS taken in 2012 or after. The socioeconomic indices were originally collected from 1996 to 2018.

**Were any ethical review processes conducted (e.g., by an institutional review board)?** If so, please provide a description of these review processes, including the outcomes, as well as a link or other access point to any supporting documentation.

No. Since LocBench is constructed based on multiple existing open-sourced datasets, we do not conduct an ethical review.

**Did you collect the data from the individuals in question directly, or obtain it via third parties or other sources (e.g., websites)?**

No. As described above, the data was collected from other open-sourced datasets.

**Were the individuals in question notified about the data collection?** If so, please describe (or show with screenshots or other information) how notice was provided, and provide a link or other access point to, or otherwise reproduce, the exact language of the notification itself.

N/A.

**Did the individuals in question consent to the collection and use of their data?** If so, please describe (or show with screenshots or other information) how consent was requested and provided, and provide a link or other access point to, or otherwise reproduce, the exact language to which the individuals consented.

N/A.

**If consent was obtained, were the consenting individuals provided with a mechanism to revoke their consent in the future or for certain uses?** If so, please provide a description, as well as a link or other access point to the mechanism (if appropriate).

N/A.

**Has an analysis of the potential impact of the dataset and its use on data subjects (e.g., a data protection impact analysis) been conducted?** If so, please provide a description of this analysis, including the outcomes, as well as a link or other access point to any supporting documentation.

204 N/A.

**Any other comments?**

206 None.

## 4 Preprocessing/cleaning/labeling

**Was any preprocessing/cleaning/labeling of the data done (e.g., discretization or bucketing, tokenization, part-of-speech tagging, SIFT feature extraction, removal of instances, processing of missing values)?** If so, please provide a description. If not, you may skip the remaining questions in this section.

We deleted the instances with missing values in the four image regression datasets from MOSAIKS [11]. We split these four datasets into training and testing sets with a ratio of 8:2. In addition, we also did log transformation for forest cover ratio, population density, and nightlight luminosity, as described above.

For both image classification and image regression datasets, we extract the image embedding of each image and directly feed them to the neural network instead of the raw images by following the practice of [6, 7, 9]. By doing that, we can speed up the location encoder training process and allow us to focus more on location encoder model development.

**Was the "raw" data saved in addition to the preprocessed/cleaned/labeled data (e.g., to support unanticipated future uses)?** If so, please provide a link or other access point to the "raw" data.

In the dataset, we exclusively provide the pre-trained image embeddings to standardize the pre-trained settings across all location encoders. Additionally, comprehensive guidance to the original data sources is available on the TorchSpatial documentation at `https://torchspatial.readthedocs.io/en/latest/`.

**Is the software that was used to preprocess/clean/label the data available?** If so, please provide a link or other access point.

No, we did not utilize any software for data preprocessing, cleaning, or labeling. The codes for data preprocessing and the split of the regression dataset can be found under the "`pre_process`" folder on GitHub at `https://github.com/seai-lab/TorchSpatial`.

**Any other comments?**

232 None.

## 5 Uses

**Has the dataset been used for any tasks already?** If so, please provide a description.

**BirdSnap**, **BirdSnap†**, **NABirds†**, **iNat2017**, and **iNat2018** have been used for geo-aware fine-grained species recognition [9]. **YFCC** has been used for Flickr image classification [6, 9], and **fMoW** for remote sensing image classification [9, 8]. **Population Density**, **Forest Cover**, **Nightlight Luminosity**, **Elevation**, **Asset Wealth Index**, **Child Mortality Rate**, **Women BMI**, **Women Edu**, **Water Index**, and **Sanitation Index** have been employed for image regression tasks. All 17 datasets are constructed and used for geo-aware tasks.

**Is there a repository that links to any or all papers or systems that use the dataset?** If so, please provide a link or other access point.

Please refer to our GitHub repository `https://github.com/seai-lab/TorchSpatial`.

**What (other) tasks could the dataset be used for?**

Other than using the datasets in LocBench to train location encoders in a supervised learning manner, we can also use all 17 datasets in LocBench for location encoder unsupervised pre-training and then adapt the model for other geo-aware tasks such as spatial interpolation, precipitation prediction, etc.

**Is there anything about the composition of the dataset or the way it was collected and preprocessed/cleaned/labeled that might impact future uses?** For example, is there anything that a dataset

consumer might need to know to avoid uses that could result in unfair treatment of individuals or groups (e.g., stereotyping, quality of service issues) or other risks or harms (e.g., legal risks, financial harms)? If so, please provide a description. Is there anything a dataset consumer could do to mitigate these risks or harms?

The datasets might be geo-biased, and inappropriate usage could enhance this bias. For example, although **iNat2017**, **iNat2018**, and **fMoW** contain images from all over the world, images are unevenly distributed: Asia, Africa, and South America are less represented.

**Are there tasks for which the dataset should not be used?** If so, please provide a description.

The datasets contained in LocBench are constructed to support the development of location encoders. They should not be used to predict sensitive indices of places such as average attractiveness, likeability, or intelligence of residents of specific places.

**Any other comments?**

None.

## 6   Distribution

**Will the dataset be distributed to third parties outside of the entity (e.g., company, institution, organization) on behalf of which the dataset was created?** If so, please provide a description.

Yes, the dataset is publicly available on the internet through this permanent figshare repository (`https://doi.org/10.6084/m9.figshare.26026798`) as well as this Dropbox link (`https://www.dropbox.com/scl/fo/lsvb50zszhup2hylphdxc/AF84XwmulxVnLYJoouq_i_Q?rlkey=tc53scmvc48di52z1k9azzymk&st=ijkms1i1&dl=0`).

How will the dataset will be distributed (e.g., tarball on website, API, GitHub)? Does the dataset have a digital object identifier (DOI)?

The dataset is distributed at figshare and the DOI of the dataset is `https://doi.org/10.6084/m9.figshare.26026798`. The dataset can also be downloaded from Dropbox (`https://www.dropbox.com/scl/fo/lsvb50zszhup2hylphdxc/AF84XwmulxVnLYJoouq_i_Q?rlkey=tc53scmvc48di52z1k9azzymk&st=ijkms1i1&dl=0`).

**When will the dataset be distributed?**

The dataset was first released in June 2024.

**Will the dataset be distributed under a copyright or other intellectual property (IP) license, and/or under applicable terms of use (ToU)?** If so, please describe this license and/or ToU, and provide a link or other access point to, or otherwise reproduce, any relevant licensing terms or ToU, as well as any fees associated with these restrictions.

All data products created through our work that are not covered under upstream licensing agreements are available via a CC BY-NC 4.0 license. To view a copy of this license, visit http://creativecommons.org/licenses/by/4.0/. All upstream data use restrictions take precedence over this license.

**Have any third parties imposed IP-based or other restrictions on the data associated with the instances?** If so, please describe these restrictions, and provide a link or other access point to, or otherwise reproduce, any relevant licensing terms, as well as any fees associated with these restrictions.

No.

**Do any export controls or other regulatory restrictions apply to the dataset or to individual instances?** If so, please describe these restrictions, and provide a link or other access point to, or otherwise reproduce, any supporting documentation.

No.

**Any other comments?**

None

## 7   Maintenance

**Who will be supporting/hosting/maintaining the dataset?**

The Spatially Explicit Artificial Intelligence Lab, led by Dr. Gengchen Mai, is responsible for supporting and maintaining the dataset.

**How can the owner/curator/manager of the dataset be contacted (e.g., email address)?**

The managers of the dataset, Nemin Wu, Qian Cao, and Gengchen Mai, can be contacted at `nemin.wu@uga.edu`, `qian.cao1@uga.edu`, and `gengchen.mai@austin.utexas.edu`.

**Is there an erratum?** If so, please provide a link or other access point.

No, it is the first release. Updates would be listed on the TorchSpatial web page (`https://github.com/seai-lab/TorchSpatial`).

**Will the dataset be updated (e.g., to correct labeling errors, add new instances, delete instances)?** If so, please describe how often, by whom, and how updates will be communicated to dataset consumers (e.g., mailing list, GitHub)?

This will be posted on the TorchSpatial web page (`https://github.com/seai-lab/TorchSpatial`).

**If the dataset relates to people, are there applicable limits on the retention of the data associated with the instances (e.g., were the individuals in question told that their data would be retained for a fixed period of time and then deleted)?** If so, please describe these limits and explain how they will be enforced.

N/A.

**Will older versions of the dataset continue to be supported/hosted/maintained?** If so, please describe how. If not, please describe how its obsolescence will be communicated to dataset consumers.

Older versions will be kept around for consistency. Details will be posted on the TorchSpatial web page (`https://github.com/seai-lab/TorchSpatial`).

**If others want to extend/augment/build on/contribute to the dataset, is there a mechanism for them to do so?** If so, please provide a description. Will these contributions be validated/verified? If so, please describe how. If not, why not? Is there a process for communicating/distributing these contributions to dataset consumers? If so, please provide a description.

Contributions from others who wish to extend, augment, or build on the dataset are welcomed. Interested contributors are encouraged to contact the corresponding author, Dr. Gengchen Mai, at `gengchen.mai@austin.utexas.edu` to discuss incorporating fixes and extensions. Contributions will be reviewed and validated by the original authors to ensure they meet the dataset's quality standards. Once validated, these contributions will be communicated and distributed to dataset consumers through existing sources. They will be distributed via the existing links and sources.

**Any other comments?**

None.

## 8   Machine Learning Reproducibility Checklist

We adopt the Machine Learning Reproducibility Checklist v2.0, released on Apr.7 2020, and exclude the contents for models, algorithms, and theoretical claim because they are not applicable for this project.

**For all datasets used, check if you include:**

- ✔ The relevant statistics, such as a number of examples.
    Yes. The number of examples can be seen in Table 1
- ✔ The details of train / validation / test splits. An explanation of any data that were excluded, and all pre-processing steps.
    Yes. We explain this information in Section 4.

✓ A link to a downloadable version of the dataset or simulation environment.

Yes. Users can access the datasets via figshare (`https://doi.org/10.6084/m9.figshare.26026798`) and Dropbox (`https://www.dropbox.com/scl/fo/lsvb50zszhup2hylphdxc/AF84XwmulxVnLYJoouq_i_Q?rlkey=tc53scmvc48di52z1k9azzymk&st=ijkms1i1&dl=0`).

☐ For new data collected, a complete description of the data collection process, such as instructions to annotators and methods for quality control.

Not applicable.

**For all shared code related to this work, check if you include:**

✓ Specification of dependencies.

Yes. See the TorchSpatial web page (`https://github.com/seai-lab/TorchSpatial`).

✓ Training code.

Yes. Same as above.

✓ Evaluation code.

Yes. Same as above.

✓ (Pre-)trained model(s).

Yes. Same as above.

✓ README file includes table of results accompanied by precise command to run to produce those results.

Yes. The tables of results are shown in README file, but the corresponding commands are in bash shell files. All of them can be found on our TorchSpatial web page (`https://github.com/seai-lab/TorchSpatial`).

**For all reported experimental results, check if you include:**

✓ The range of hyper-parameters considered, method to select the best hyper-parameter configuration, and specification of all hyper-parameters used to generate results.

Yes. See Appendix Section 5.

✓ The exact number of training and evaluation runs.

Yes. See README file on our TorchSpatial web page (`https://github.com/seai-lab/TorchSpatial`).

✓ A clear definition of the specific measure or statistics used to report results.

Yes. See paper Section 3.3 and Appendix Section 4.

✗ A description of results with central tendency (e.g. mean) and variation (e.g. error bars).

We provide experimental results of 15 location encoders on all datasets contained in LocBench. Reporting central tendency and variation for all of them is prohibitively expensive. We will provide them in the future through our website.

✓ The average runtime for each result, or estimated energy cost.

Yes. See README file on our TorchSpatial web page (`https://github.com/seai-lab/TorchSpatial`).

✓ A description of the computing infrastructure used.

Yes. See Appendix Section 5.