# OpenReview forum: "TorchSpatial: A Location Encoding Framework and Benchmark for Spatial Representation Learning"
_NeurIPS.cc/2024/Datasets_and_Benchmarks_Track — NeurIPS 2024 Track Datasets and Benchmarks Poster_

### Official Review · Reviewer_xzSd · 2024-07-20
**A solid paper with some flaws**

**Rating:** 7
**Confidence:** 4
**Clarity:** The paper is well written.

**Review:**

Quality: The quality of the article's figures and arguments is high, with a certain degree of persuasiveness and innovation, but it is not deep enough.

Clarity: The organization of the article is clear and the writing is smooth, but many important details are placed in the appendix, which may not be appropriate.

Originality: The framework and benchmarks proposed in the article mainly integrate previous work, with limited innovation. However, the method for quantifying geographic bias proposed by the author has good originality.

Significance: TorchSpatial offers a unified and highly extensible framework and benchmark for the research in SRL, which can assist in propelling the development of downstream tasks such as Geospatial Artificial Intelligence (GeoAI).

Pros:

1. The dataset in the benchmark is comprehensive and covers the main regression and classification tasks in SRL.

2. The location encoding has a wide range of applicability and takes into account the encoding of various spatial data types.

3. The proposed novel Geo-Bias Score is able to quantify geographic bias more effectively than existing metrics, while having better interpretability.

Cons:

1. The authors have the interaction of image encoding and location encoding appearing in the model diagrams, but this doesn't seem to be presented in the paper. In addition, the introduction of location encoding is too brief.

2. The description of Geo-bias lacks the necessary mathematical formulas.

3. Geo-Bias Score has some hyperparameters that are not easy to set, which may affect its application.

**Strengths:**

1. The framework and benchmarks proposed by the author are comprehensive, taking into account various scenarios.

2. The author has noted the impact of geographical bias and innovatively proposed Geo-bias to effectively quantify this bias, achieving an assessment of geographical bias for geo-aware AI models.

3. The article has a clear structure that allows for smooth reading; the extensive experiments are sufficient to support the arguments presented in the article.

**Additional Feedback:**

None.

**Correctness:**

The framework and benchmarks are reasonably well constructed and source code is also provided.

**Documentation:**

The documentation is accessible on Github.

**Ethics:**

None.

**Limitations:**

The authors discuss limitations in Section 5.

**Opportunities For Improvement:**

1. The mathematical formulation of Geo-Bias Score can be provided and describe the choice of hyperparameters for different scenarios.

2. The description of location encoding could be a bit richer, while its relation to image encoding could be briefly described.

**Relation To Prior Work:**

The relationship to previous work is clearly discussed.

**Summary And Contributions:**

This paper introduces a deep learning framework and benchmark TorchSpatial for positional encoding which is a key component of spatial representation learning(SRL). TorchSpatial consists of three parts: 1) It proposes a unified framework that includes 15 commonly used positional encodings, capable of transforming geographical location data into a format that is easy for neural networks to process, ensuring the scalability and reproducibility of the implementation. 2) It constructs a benchmark, LocBench, which includes 7 geo-aware image classification datasets and 4 geo-aware image regression datasets. 3) It introduces a novel method to quantify geographic bias, the Geo-Bias Score. The authors have conducted a series of experiments to demonstrate that TorchSpatial can fill the gap in SRL in terms of a unified community-shared framework and benchmark.

---

> ### Author Rebuttal · Authors · 2024-08-17
>
> ## 1. Lack of interaction between image encoding and location encoding in the paper. In addition, the introduction of location encoding is too brief.
> We have added additional pseudocode for the model architecture of both regression and classification tasks in TorchSpatial. Please refer to our general response for details.
>
> ## 2. Geo-bias lacks the necessary mathematical formulas.
> Geo-bias is computed based on the Spatial Self-Information (SSI) proposed in [1]. Intuitively, it computes the negative log probability of observing a certain spatial pattern of data under the hypothesis thatwhere the data appear in the space is completely random. The larger the value, the more non-trivial the pattern is, thus indicating the data are spatially biased. Briefly speaking, [1] proved that such probability distribution is approximately Gaussian with mean $\mu = \sum_{p \neq q} (c_p - \bar{x})(c_q - \bar{x})\mu_{p,q} + \sum_{p}(c_p - \bar{x})^2\mu_{p,p}$ and variance $\sigma^2 = \sum_{p \neq q \neq r_{max}} [(c_p - \bar{x})(c_q - \bar{x}) - 2(c_p - \bar{x})(c_{r_{max}} - \bar{x}) + (c_{r_{max}} - \bar{x})^2]^2 \sigma^2_{p,q} + \sum_{p \neq r_{max}}[(c_p - c_{r_{max}})^2]^2 \sigma_{p,p}$. For the detailed explanations on how to compute these terms, please refer to [1] Section 3.5, Theorem 8 and Theorem 9 for full mathematical formulas.
>
> In addition, we have added pseudocode in the general response to briefly represent the calculation process of geo-bias scores.
>
> [1] Zhangyu Wang, Krzysztof Janowicz, Gengchen Mai, and Ivan Majic. Probing the information theoretical roots of spatial dependence measures, 2024.544
>
> ## 3. Geo-Bias Score has some hyperparameters that are not easy to set.
> The setting of hyperparameters (k, r) can be based on  prior knowledge, i.e.,  domain knowledge mandates certain selection of hyperparameters. For example, in ecology, k may refer to observations of wild lifes and r may refer to the migration range, which from the ecologists’ expertise we know the best choice of k and r.
>
> In case the users do not have domain knowledge for hyperparameter setting, we have the following rule of thumb(not necessarily optimal but statistically stable) strategy:
> * take the dataset, select k larger than 30 (to ensure statistical significance; we recommend around 100), and
> * compute the k-nearest neighbors of each data point. Then, compute the average distance of the farthest neighbor and use it as r.
> This is also how we selected the hyperparameters for our reported experiments and it proves working fairly well.
>
> As for B, we simply choose the number that makes the average distance among the background points similar to that of the observed points.

---

> > ### Comment · Reviewer_xzSd · 2024-08-26
> >
> > Thank you for your response. Your reply has addressed my queries regarding the Location Encoder and the Geo-bias Score. Consequently, I will maintain the current score.

---

### Official Review · Reviewer_rFTf · 2024-07-22
**Great work, but I have some concerns about the dataset license**

**Rating:** 7
**Confidence:** 5
**Clarity:** Yes, it is.

**Review:**

Quality and clarity: Very good paper, well written and easy to follow. The experimental setup sounds correct and authors have taken care of every detail.

Significance: As the authors claim, this would be the first location encoding framework open to the research community.

Pros: New framework, new geo-bias score.
Cons: I am quite worried the authors may be opening the datasets under a different license the original authors used for theirs. fMOW and YFCC include imagery that cannot be used for commercial use. In the supplementary material, section 6, the authors release the datasets under MIT license. Moreover, the authors provide a link where the datasets are hosted. I wonder if the authors got the approval to host and share the datasets.

**Strengths:**

Two main contributions: a novel framework and new geo-bias score metrics.

**Additional Feedback:**

I do think I already provided some suggestions.

**Correctness:**

The dataset is available through the links the authors share in the supplementary material. The experimental setup is well conducted and appropriate.

**Documentation:**

My only concern at this point is the licensing part. Maintenance plan looks good as well as benchmark.

**Ethics:**

Again, the license type and share of the datasets need to be clarified.

**Limitations:**

As a I mention earlier, the authors need to clarify the type of license they intend to use and hosting.

**Opportunities For Improvement:**

I think that the observation they mention in 294-298 should be further investigated. It would be a great addition to show some cases over the entire globe in terms of bias.

**Relation To Prior Work:**

Previous work is well described and how their work differs from previous ones.

**Summary And Contributions:**

The authors present a new location encoding framework and benchmark, simple and easy to use. They build their framework around several datasets, conducting different tasks. They additionally present a new geo-bias score (base and global) based on Moran's I statistics, and conduct other metrics to assess different location encoders.

---

> ### Author Rebuttal · Authors · 2024-08-17
>
> ## 1. Potential license conflict.
>
> Your careful attention to the licensing considerations is both thorough and greatly appreciated. We would like to clarify that the fMOW and YFCC datasets we are releasing do not contain the original imagery but rather the image embeddings learned through an image encoder.
> However, your concern is well-founded. We have taken additional precautions in our latest dataset release. The table below provides information on the licenses for the datasets we used.
> Dataset|License
> -|-
> BirdSnap|unknown
> NABirds|customized license (Education use only)
> BirdSnap†|unknown
> NABirds†|unknown
> iNat2017|MIT License
> iNat2018|MIT License
> YFCC100M-GEO100 dataset|unknown (It is derived from YFCC100M, which has no claims on Adapted Material.)
> fMoW|customized license (Non-commercial use only)
> SustainBench|MIT License
> MOSAIKS|CC BY 4.0 license
>
> We will change the license into CC BY-NC 4.0 and claim all upstream data use restrictions take precedence over this license.
>
> ## 2.  The observation in 294-298 should be further investigated. It would be a great addition to show some cases over the entire globe in terms of bias.
>
> Your input is very insightful and thought-provoking. As you observed, Table 4 shows that adding location encoders in some cases will increase the model’s geographic bias but not as much as we see in Table 2. This is mainly because the data points from 4 regression datasets are uniformly-at-random (UAR) sampled from the globe.
>
> The difference in sampling strategy can significantly impact performances of SRL. In our case, the datasets in Table 2 are unevenly distributed as illustrated in Figure 3 and Figure 4, while data for regression are uniformly-at-random (UAR) sampled from the globe, which is more informative and more learning-friendly for location encoders. To better understand these effects, we are working on adding a controlled experiment to see whether uniform sampling and Gaussian probability sampling will influence the geo-bias score in Table 4. The experiments are still ongoing. The result table and further explanation will be added to the Appendix of the camera-ready version.
>
> We also appreciate your suggestion that shows some cases over the entire globe in terms of bias. We have implemented hot spot analysis on Hit@1 of three models on three datasets to reveal the geographic bias across the globe. Please refer to Figure 6 in Appendix.

---

> > ### Comment · Reviewer_rFTf · 2024-08-20
> > **Recommendation for Acceptance**
> >
> > Considering that the license has been corrected as well as the redistrubution clarification, I do recommend the paper for acceptance.

---

### Official Review · Reviewer_ePbQ · 2024-08-08
**A Comprehensive Framework for Enhancing Location Encoding in Geospatial AI**

**Rating:** 6
**Confidence:** 4
**Correctness:** Yes
**Clarity:** Yes

**Review:**

**Pros:**

The introduction of TorchSpatial is a pioneering effort, addressing a critical void in current GeoAI studies by offering a cohesive location encoding framework.

Incorporating 15 widely recognized location encoders ensures a broad and inclusive approach, improving scalability and reproducibility.

The development of LocBench, featuring 7 geo-aware image classification and 4 geo-aware image regression datasets, provides a substantial benchmark for evaluating SRL models.

The variety of datasets ensures the benchmark encompasses a wide spectrum of applications, making it pertinent to numerous geospatial tasks.

Creating a detailed suite of evaluation metrics, including the groundbreaking Geo-Bias Score, not only evaluates overall model performance but also addresses the essential issue of geographic bias, promoting equity in GeoAI research.

The paper meticulously analyzes model performance and geographic bias, offering significant insights into the advantages and drawbacks of various location encoders.

This analysis can direct future research and innovation in SRL, encouraging progress in the field.

Making the TorchSpatial model framework, LocBench, and Geo-Bias Score evaluation framework available to the public guarantees transparency and encourages extensive use and collaboration within the research community.

**Cons:**

Pseudocode for key components is not sufficient which could aid researchers in better understanding the framework.

Including practical case studies or real-world applications demonstrating TorchSpatial's utility in addressing geospatial problems would make the paper more compelling to a broader audience.

**Strengths:**

By providing tools to assess and mitigate bias, TorchSpatial encourages researchers to consider the ethical implications of their work, promoting the development of more socially responsible geospatial technologies.

**Additional Feedback:**

Detailed explanations of the framework's architecture and workflows would enhance understanding. Practical case studies or real-world applications would illustrate the framework's utility.

**Documentation:**

Yes

**Ethics:**

No ethical issues found

**Limitations:**

**Suggestions:**

The authors can mention any constraints related to LocBench datasets, such as potential biases in the datasets themselves or limitations in their geographic or temporal coverage. This can help users understand the contexts in which the benchmark is most applicable.

While the introduction of the Geo-Bias Score is commendable, the authors can also discuss how biases in the training data can still propagate through models, despite this metric, emphasizing the importance of using diverse and representative datasets.

**Opportunities For Improvement:**

Although the analysis of model performance and geographic bias is extensive, incorporating a broader range of experiments and comparative studies with existing SRL models would add depth.

Including more quantitative results, such as performance metrics and bias scores for each encoder, would provide a clearer picture of the framework's effectiveness.

Expanding on the verification process of LocBench datasets would fortify the paper. Including details the methodology used to ensure the datasets' relevance and quality in the Collection Section in Appendix would be beneficial.

**Relation To Prior Work:**

Yes

**Summary And Contributions:**

The paper presents TorchSpatial, an innovative framework designed to enhance spatial representation learning (SRL) by focusing on location encoding. TorchSpatial offers a unified approach to location encoding, a robust benchmark (LocBench), and a novel suite of evaluation metrics, including the Geo-Bias Score.

The framework delivers an extensive analysis and insights into the performance and geographic bias of various location encoders, guiding future research and development. Furthermore, by making TorchSpatial, LocBench, and the Geo-Bias Score evaluation framework publicly available, the paper ensures transparency and encourages collaboration within the research community.

---

> ### Author Rebuttal · Authors · 2024-08-17
>
> ## 1.  Pseudocode for key components is not sufficient.
>
> We have added pseudocode for the model architecture of both regression and classification tasks.  Please see #1 in our general response for details.
>
> ## 2. Including a broader range of experimental case studies
>
> In this paper, we conducted 7 tasks in two categories, and all of them are real-world experiments. We compared the performance of 15 existing location encoders in these experiments and put the results in Table 1 and Table 3. Additionally, comparative studies on the geographic bias of these 15 models are also implemented and readers can refer to Table 2 and Table 4 for more details.
>
> Following your suggestion, we will expand TorchSpatial framework to include 6 additional regression tasks based on datasets from SustainBench [1] – a comprehensive suite of sustainability benchmarks aligned with the UN Sustainable Development Goals (SDGs). These new tasks will involve regressing global sustainability indices, including the asset wealth index, women's BMI, water index, under-5 mortality, women educational attainment, and sanitation index. As a result, TorchSpatial will feature a total of 7 datasets for classification tasks and 10 datasets for regression tasks, offering extensive coverage of real-world challenges.
>
> We will provide detailed evaluations for all 15 location encoders, including R², MSE, MAE, and geo-bias scores. These evaluations, along with comprehensive details of the datasets, will be included in the Appendix of the main content for the camera-ready version. Due to the time constraints for the rebuttal, the experiments are still ongoing. We present a portion of the results in the following table.
>
> Table 1. The R^2 of different models on geo-aware image regression datasets in benchmark. **Bold** indicates the best models in Group B and C.
> |Model|Women_edu|SanitationIndex|
> |---------------------------------|--------------------|--------------|
> |NoPrior(i.e.image model)|0.22|0.33|
> |*tile*|0.00|0.00|
> |*wrap*|0.34|0.31|
> |*wrap+ffn*|0.38|0.32|
> |*rbf*|**0.51**|**0.48**|
> |*rff*|0.44|0.37|
> |*SpaceVec-grid*|0.35|0.33|
> |*spaceVec-theory*|0.43|0.39|
> |*xyz*|0.43|0.19|
> |*NeRF*|**0.59**|**0.53**|
> |*SphereVec-sphereC*|0.58|0.52|
> |*SphereVec-sphereC+*|**0.59**|0.46|
> |*SphereVec-sphereM*|**0.59**|0.52|
> |*SphereVec-sphereM+*|0.37|0.31|
> |*SphereVec-dfs*|0.39|0.33|
> |*Siren(SH)*|0.48|0.42|
>
> Please refer to the Official Comment for the table of MRR of different models on 7 geo-aware image classification datasets, and two more tables of MSE and MAE of different models on geo-aware image regression datasets in benchmark.
>
>
> [1] Yeh, Christopher, et al. "Sustainbench: Benchmarks for monitoring the sustainable development goals with machine learning." arXiv preprint arXiv:2111.04724 (2021)
>
> ## 3. Including more quantitative results, such as performance metrics and bias scores for each encoder.
> For classification tasks, other than the accuracy score we reported in the paper, we additionally calculated the MRR and Top-3 Accuracy for all 15 models across 7 datasets. For regression tasks, we provided additional tables with MSE and MAE values. Additionally, we analyzed geo-bias by evaluating Hit@3 across different geographic regions.
>
> Two  examples of table is in the attached PDF for your reference. Other tables are not able to show here due to word limit. We will include these additional metrics in the Appendix in the final version to offer a more comprehensive evaluation of our models. A more detailed explanation of geo-bias scores (see #2 in the General Response Section) will also be added to clarify their interpretation and impact for camera-ready version.
>
> ## 4. Expanding on the verification process of LocBench datasets, including details of the methodology used to ensure the datasets' relevance and quality in the Collection Section in the Appendix.
> The collection process is depicted in Section 3 Collection Process on Supplementary Material.
>
> For the relevance of these 11 datasets, they are specifically chosen because they have a wide range of spatial distribution, which can demonstrate the effectiveness of location encoders on the corresponding tasks. Most of them are globally distributed, while only NABird† and YFCC have relatively smaller ranges of spatial distribution, one for America and the other for the United States.
>
> We verify the datasets’ quality by following operations:
> * Verify dataset information with original supplier catalogs. No inconsistency or discrepancy between the dataset and trusted sources is found.
> * Data cleaning and data transformation for 4 geo-aware image regression datasets. Please refer to lines 158 to 187 for more details.
> * Assess the datasets for completeness and coverage. We plot the geographic coverage of each dataset in Figure 3, Figure 4, and Figure 5, and all of them have a comprehensive geographical scope and cover major regions in the world.
>
> We will expand the Collection Process section with a more detailed verification process in the camera-ready version.

---

> > ### Author Response · Authors · 2024-08-17
> > **Tables of evaluation metrics for classification and regression tasks**
> >
> > We put three tables here due to the character limit in Rebuttal as a part of the response to #2 "Including more quantitative results".
> >
> > Table 1. The MRR of different models on 7 geo-aware image classification datasets in LocBench. **Bold** indicates the best models in Group B and C. "Avg" is the average performance on all species recognition datasets.
> >
> > |Model|BirdSnap|BirdSnap†|NABirds†|iNat2017|iNat2018|Avg|YFCC|fMOW|
> > |------------------------------|----------|-----------|----------|----------|----------|-------|-------|-------|
> > |NoPrior(i.e.imagemodel)|0.790|0.790|0.841|0.728|0.705|0.771|0.644|0.785|
> > |*tile*|0.790|0.787|0.839|0.727|0.673|0.763|0.642|0.785|
> > |*wrap*|**0.801**|0.856|**0.881**|0.764|0.807|0.822|0.651|0.790|
> > |*wrap+ffn*|**0.801**|0.856|0.877|**0.778**|0.804|0.823|0.650|0.788|
> > |*rbf*|0.786|0.861|0.880|0.767|0.735|0.806|0.654|**0.793**|
> > |*rff*|0.799|0.852|0.880|0.767|0.800|0.820|0.651|0.791|
> > |*SpaceVec-grid*|0.789|0.860|0.880|0.767|0.807|0.820|**0.656**|**0.793**|
> > |*spaceVec-theory*|0.798|**0.862**|0.880|0.767|**0.812**|**0.824**|0.655|0.791|
> > |*xyz*|0.801|0.854|0.875|0.768|0.796|0.819|0.651|0.788|
> > |*NeRF*|**0.802**|0.862|0.880|0.771|**0.809**|0.825|0.649|0.786|
> > |*SphereVec-sphereC*|**0.802**|**0.863**|**0.881**|**0.777**|0.806|**0.826**|**0.657**|0.793|
> > |*SphereVec-sphereC+*|0.801|**0.863**|0.877|0.757|**0.809**|0.821|0.656|0.792|
> > |*SphereVec-sphereM*|0.801|0.861|**0.881**|0.757|0.806|0.821|0.655|**0.795**|
> > |*SphereVec-sphereM+*|0.800|0.854|0.875|0.775|0.803|0.821|0.642|0.785|
> > |*SphereVec-dfs*|0.800|0.852|0.879|0.776|0.801|0.822|0.648|0.789|
> > |*Siren(SH)*|0.800|0.854|0.880|**0.777**|0.796|0.821|0.648|-|
> >
> >
> > Table 2. The MSE of different models on geo-aware image regression datasets in LocBench. **Bold** indicates the best models in Group B and C.
> >
> > |Model|Population Density|Forest Cover|Nightlight Luminosity|Elevation|
> > |--------------------------------------------|--------------------|--------------|-----------------------|--------------------|
> > |NoPrior(i.e.image model)|3.359|1.612|0.114|624190.634|
> > |*tile*|3.341|1.853|0.119|1110641.543|
> > |*wrap*|**1.455**|**1.002**|0.089|153042.230|
> > |*wrap+ffn*|2.133|1.239|0.097|189815.589|
> > |*rbf*|2.631|1.641|0.090|495162.168|
> > |*rff*|1.979|1.014|0.100|166058.184|
> > |*SpaceVec-grid*|1.882|1.030|0.114|158759.246|
> > |*spaceVec-theory*|2.554|1.124|**0.077**|**152374.962**|
> > |*xyz*|3.118|1.399|0.121|217648.783|
> > |*NeRF*|2.712|1.174|0.105|170574.877|
> > |*SphereVec-sphereC*|2.165|**0.935**|0.098|137770.593|
> > |*SphereVec-sphereC+*|2.327|0.973|0.099|**117792.722**|
> > |*SphereVec-sphereM*|1.323|1.017|0.107|123583.103|
> > |*SphereVec-sphereM+*|1.643|1.193|0.084|186836.893|
> > |*SphereVec-dfs*|1.756|1.154|0.114|238719.098|
> > |*Siren(SH)*|**1.241**|0.942|**0.083**|140656.674|
> >
> >
> > Table 3. The MAE of different models on geo-aware image regression datasets in LocBench. **Bold** indicates the best models in Group B and C.
> >
> > |Model|PopulationDensity|ForestCover|NightlightLuminosity|Elevation|
> > |------------------------------|--------------------|--------------|-----------------------|-----------|
> > |NoPrior(i.e.imagemodel)|1.471|0.935|0.107|495.021|
> > |*tile*|1.473|1.037|0.124|640.146|
> > |*wrap*|**0.848**|**0.659**|0.119|261.671|
> > |*wrap+ffn*|1.090|0.787|0.106|285.729|
> > |*rbf*|1.269|0.952|**0.091**|427.984|
> > |*rff*|1.057|0.683|0.116|266.039|
> > |*SpaceVec-grid*|1.018|0.700|0.120|251.110|
> > |*spaceVec-theory*|1.227|0.711|0.101|**234.266**|
> > |*xyz*|1.382|0.877|0.125|312.483|
> > |*NeRF*|1.233|0.760|0.103|256.976|
> > |*SphereVec-sphereC*|1.116|**0.635**|0.099|237.501|
> > |*SphereVec-sphereC+*|1.149|0.639|0.101|220.088|
> > |*SphereVec-sphereM*|0.799|0.668|0.101|**215.829**|
> > |*SphereVec-sphereM+*|0.904|0.763|0.098|284.692|
> > |*SphereVec-dfs*|0.952|0.750|0.136|311.222|
> > |*Siren(SH)*|**0.771**|**0.635**|**0.093**|238.202|

---

### Author Rebuttal · Authors · 2024-08-17

## 1. Provide a more detailed Location Encoder description

Both reviewers ePbQ and xzSd recommended providing a more detailed description of the model architecture in TorchSpatial. In response, we have included an additional pseudocode illustrating the model architecture for both regression and classification tasks. This content will be included in the camera-ready version. Additionally, we will revise the introduction to location encoding for greater clarity.

As outlined in the pseudocode in the attached PDF document:

For classification tasks: Inspired by [1], we can include the location information as a Bayesian spatio-temporal prior. Our model classifies the image and location data separately using an image classifier and a location classifier respectively. The final prediction is obtained by performing an element-wise multiplication of the results from these two classifiers. In this way, after pretraining the image encoder, we control and fix both the image encoders and image embeddings to focus on the performance of different location encoders.

For regression tasks: The model uses a more direct way to extract features (embeddings) from both the image and location data using encoders, concatenates these features, and then utilizes an MLP to predict a continuous value based on the combined information.

[1] Mac Aodha, Oisin, Elijah Cole, and Pietro Perona. "Presence-only geographical priors for fine-grained image classification." Proceedings of the IEEE/CVF International Conference on Computer Vision. 2019.


## 2. Provide a more detailed Geo-bias Scores description

Given a list of locations (l_1, l_2, … l_n) and a list of corresponding performance values (p_1, p_2, … p_n), for classification tasks, p_i is binary (either “High” or “Low”); for regression tasks, for the unified geo-bias formulation and computational stability, we use a threshold to turn the real-valued performance score into binary “High” and “Low”. The selection of the threshold depends on specific tasks. Hyperparameter k, the maximum number of observations in the neighborhood considered in the geo-bias score computation; hyperparameter r, the maximum radius of neighborhood considered in the geo-bias score computation; hyperparameter B, the number of background points sampled for each location.

Implementation Procedure:

1. For each l_i find its nearest k neighbors (l_i1, l_i2, … l_ik) that fall into the circle of radius r, called neighborhood.
2. Randomly sample B background points (b_1, b_2, … b_B) within the neighborhood, setting their values to be all 0s.
3. Base geo-bias score: set the performance values of (l_i1, l_i2, … l_ik) to be all 1s, mix up with the background points, and compute the spatial self-information (SSI) for this set of points.
4. Relative geo-bias score: set the performance values of (l_i1, l_i2, … l_ik) to be 1 for high performance and -1 for low performance. Mix up with the background points, and compute the spatial self-information (SSI) for this set of points.

---

### Decision · Program_Chairs · 2024-09-26

**Decision:**

Accept (Poster)

**Comment:**

The authors propose TorchSpatial, a learning framework and benchmark· for location (point) encoding for spatial representation learning. This paper was a clean acceptance and will have a high impact on the community. The reviewers primarily liked the paper, and the discussion during the rebuttal was meaningful, engaging, and thought-provoking.
I will note that the comment from one reviewer on the licensing issue was carefully addressed during the rebuttal.